# RÉNYI NEURAL PROCESSES

## ABSTRACT

Neural Processes (NPs) are deep probabilistic models that represent stochastic processes by conditioning their prior distributions on a set of context points. Despite their obvious advantages in uncertainty estimation for complex distributions, NPs enforce parameterization coupling between the conditional prior model and the posterior model, thereby risking introducing a misspecified prior distribution. We hereby revisit the NP objectives and propose Rényi Neural Processes (RNP) to ameliorate the impacts of prior misspecification by optimizing an alternative posterior that achieves better marginal likelihood. More specifically, by replacing the standard KL divergence with the Rényi divergence between the model posterior and the true posterior, we scale the density ratio $\frac{p}{q}$ by the power of (1-$\alpha$) in the divergence gradients with respect to the posterior. This hyper parameter $\alpha$ allows us to dampen the effects of the misspecified prior for the posterior update, which has been shown to effectively avoid oversmoothed predictions and improve the expressiveness of the posterior model. Our extensive experiments show consistent log-likelihood improvements over state-of-the-art NP family models which adopt both the variational inference or maximum likelihood estimation objectives. We validate the effectiveness of our approach across multiple benchmarks including regression and image inpainting tasks, and show significant performance improvements of RNPs in real-world regression problems where the underlying prior model is misspecifed.

## 1 INTRODUCTION

Neural processes (NPs) (Garnelo et al., 2018b) strive to represent stochastic processes via deep neural networks with desirable properties in uncertainty estimation and flexible feature representation. The vanilla NP (Garnelo et al., 2018b) predicts the distribution for unlabelled data given any set of observational data as *context*. The main advantage of NPs is to learn a set-dependent prior distribution, where the KL divergence is minimized between a posterior distribution conditioned on a *target* set with new data and the prior distribution conditioned on the context set (Kim et al., 2019; Jha et al., 2022; Bruinsma et al., 2023). However, as the parameters of such conditional prior are unknown, NP proposes a coupling scheme where the prior model that parameterizes the distribution is forced to share its parameters with an approximate posterior model. As the model parameter space induced by neural networks is usually large, enforcing coupling on such a space could lead to prior misspecification which could consequently produce a biased estimate of the posterior variance and deteriorate predictive performance (Cannon et al., 2022; Knoblauch et al., 2019). Such misspecification can be worsened under noisy context set (Jung et al., 2024; Liu et al., 2024). Other cases of prior misspecification encompass domain shifts (Xiao et al., 2021), out-of-distribution predictions (Malinin & Gales, 2018) and adversarial samples (Stutz et al., 2019).

To address the prior misspecification caused by parameterization coupling in vanilla NPs, several studies have been proposed to relax the constraint (Wang et al., 2023; Wang & Van Hoof, 2022; Wu et al., 2018; Wicker et al., 2021). For instance, the prior and the posterior models can share partial parameters instead of the entire network (Rybkin et al., 2021; Liu et al., 2022); hierarchical latent variable models are also utilized where both models share the same global latent variable and induce prior or posterior-specific distribution parameterization (Shen et al., 2023; Requeima et al., 2019; Kim et al., 2021; Lin et al., 2021). In this paper, we offer a new insight of handling prior misspecification in NPs through the lens of robust divergence (Futami et al., 2018), which seeks to learn an alternative posterior without changing the parameters of interests. Instead of minimizing

the standard KL divergence between the prior and posterior distributions, robust divergences are theoretically guaranteed to produce better posterior estimates under prior misspecification (Verine et al., 2024; Regli & Silva, 2018). The Rényi divergence (Li & Turner, 2016; Van Erven & Harremos, 2014b), for instance, introduces an additional parameter $\alpha$ to control how the prior distribution can regularize the posterior during the posterior updates. By scaling the density ratio $\frac{p}{q}$ by the power of $(1-\alpha)$ in the divergence gradients with respect to the posterior, this hyper parameter $\alpha$ allows us to reduce the regularization effects of the misspecified prior (Knoblauch et al., 2019), thereby mitigating the biased estimates of the posterior variance, avoiding oversmoothed predictions (Alemi et al., 2018; Higgins et al., 2017), and achieving performance improvements. Due to the constraint of parameterization coupling in NPs, such settings can give rise to a potentially critical case of prior misspecification and the need for more robust divergences in NP learning.

In light of this, we propose Rényi Neural Processes (RNPs) that focus on improving neural processes with a more robust objective. RNP minimizes the Rényi divergence between the posterior distribution defined on the target set and the true posterior distribution given the context and target sets. We prove that RNP connects the common variational inference and maximum likelihood estimation objectives for training vanilla NPs via the hyperparameter $\alpha$, through which RNP provides the flexibility to dampen the effect of the misspecified prior and empower the posterior model for better expressiveness. Our main contributions are summarized as:

1. We introduce a new objective for neural processes that unifies the variational inference and maximum likelihood estimation objectives via the Rényi divergence.

2. We show that Rényi neural processes can be applied to several state-of-the-art neural process family models in a simple yet effective manner without changing the model. We validate the effectiveness of Rényi neural processes on comprehensive experiments including regression, image inpainting, and prior misspecification.

## 2 PRELIMINARIES

**Neural Processes**: Neural processes are a family of deep probabilistic models that represent stochastic processes (Wang & Van Hoof, 2020; Lee et al., 2020). Let $f_\tau : \mathbb{X} \to \mathbb{Y}$ be a function sampled from a stochastic process $p(f)$ where each $f_\tau$ maps some input features $\mathbf{x}$ to an output $\mathbf{y}$ and $\mathbb{D}_{\text{train}}$ and $\mathbb{D}_{\text{test}}$ are meta-tasks induced by different $f_{\text{train}}$ and $f_{\text{test}}$ during meta-training and meta-testing. For a specific task $\mathbb{D}_\tau$, we split the data further into a context set $\mathbb{C} : (X_C, Y_C) := \{(\mathbf{x}_m, \mathbf{y}_m)_{m=1}^M\}$ and a target set $\mathbb{T} : (X_T, Y_T) := \{(\mathbf{x}_n, \mathbf{y}_n)_{n=1}^N\} = \mathbb{D}_\tau \backslash \mathbb{C}$. Our goal is to predict the target labels given the target inputs and the observable context set: $p(Y_T|X_T, X_C, Y_C)$. NPs (Garnelo et al., 2018b) introduce a latent variable $\mathbf{z}$ to parameterize the conditional distribution $p(f|\mathbb{C})$ and define the model (see Fig 1) as $p(Y_T|X_T, X_C, Y_C) = \int p_\theta(Y_T|X_T, \mathbf{z})p_\varphi(\mathbf{z}|X_C, Y_C)d\mathbf{z}$ where $\theta$ and $\varphi$ are network parame-

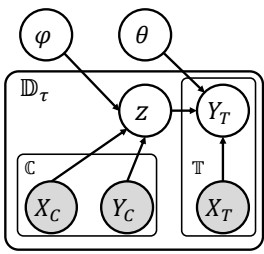

Figure 1: Graphical Model.

ters of the likelihood and prior (also known as recognition) models, respectively. Due to the intractable likelihood, two types of objectives including the variational inference (**VI**) and maximum likelihood (**ML**) estimation have been proposed to optimize the parameters (Foong et al., 2020; Nguyen & Grover, 2022; Bruinsma et al., 2023; Guo et al., 2023):

$$-\mathcal{L}_{VI}(\theta, \phi, \varphi) = \mathbb{E}_{\mathbb{D}_{\text{train}}}\left[\mathbb{E}_{q_\phi(\mathbf{z})} \log p_\theta(Y_T|X_T, \mathbf{z}) - D_{\text{KL}}\left(q_\phi(\mathbf{z})\|p_\varphi(\mathbf{z}|X_C, Y_C)\right)\right] \quad (1)$$

$$-\mathcal{L}_{ML}(\theta, \varphi) = \mathbb{E}_{\mathbb{D}_{\text{train}}}\left[\mathbb{E}_{p_\varphi(\mathbf{z}|X_C, Y_C)} \log p_\theta(Y_T|X_T, \mathbf{z})\right] \quad (2)$$

The approximate posterior distribution for VI-based methods is usually chosen as $q_\phi(\mathbf{z}) = q_\phi(\mathbf{z}|X_T, Y_T, X_C, Y_C)$. As the parameters of the conditional prior $p_\varphi(\mathbf{z}|X_C, Y_C)$ are unknown, NPs couple its parameters with the approximate posterior $p_\varphi(\mathbf{z}|X_C, Y_C) \approx q_\phi(\mathbf{z}|X_C, Y_C)$. We now replace the notation of the approximate posterior with $\varphi$ for ML consistency:

$$-\mathcal{L}_{VI}(\theta, \varphi) \approx \mathbb{E}_{\mathbb{D}_{\text{train}}}\left[\mathbb{E}_{q_\varphi(\mathbf{z})} \log p_\theta(Y_T|X_T, \mathbf{z}) - D_{\text{KL}}\left(q_\varphi(\mathbf{z}|X_T, Y_T, X_C, Y_C)\|q_\varphi(\mathbf{z}|X_C, Y_C)\right)\right] \quad (3)$$

The KL term in Eq 3 is sometimes referred to as the **consistency regularizer** (Wang et al., 2023; Foong et al., 2020), which encourages the target set to be subsumed into the context set. This

assumption, as will show later, is the source of the inference suboptimality of vanilla NPs in the existence of finite capacity/data.

**Rényi Divergences**: The Rényi divergence (**RD**) (Van Erven & Harremos, 2014a) is defined on two distributions with a hyperparamter $\alpha \in (0, +\infty)$ and $\alpha \neq 1$:

$$D_\alpha(q(\mathbf{z})\|p(\mathbf{z})) = \frac{1}{\alpha-1}\log\int q(\mathbf{z})^\alpha p(\mathbf{z})^{1-\alpha}d\mathbf{z} = \frac{1}{\alpha-1}\log\mathbb{E}_{q(\mathbf{z})}\left[\frac{p(\mathbf{z})}{q(\mathbf{z})}\right]^{1-\alpha}. \tag{4}$$

Note that the RD is closely related to the KL divergence in that if $\alpha \to 1$ then $D_\alpha(q\|p) \to D_{\mathrm{KL}}(q\|p)$ (Van Erven & Harremos, 2014a). In other words, choosing $\alpha$ close to 1 would result in a posterior as close to the standard VIs. Changing the KL divergence to RD can induce a robust posterior via the hyperparameter $\alpha$. To see this, consider the gradients of the RD wrt the posterior parameters:

$$\frac{\partial}{\partial\varphi}\mathcal{D}_\alpha\left(q_\varphi(\mathbf{z})\|p(\mathbf{z})\right) = \frac{\alpha}{\alpha-1}\frac{\int\left[\frac{p(\mathbf{z})}{q_\varphi(\mathbf{z})}\right]^{1-\alpha}\frac{\partial q_\varphi(\mathbf{z})}{\partial\varphi}d\mathbf{z}}{\mathbb{E}_{q_\varphi(\mathbf{z})}\left[\frac{p(\mathbf{z})}{q_\varphi(\mathbf{z})}\right]^{1-\alpha}} \tag{5}$$

where the influence of the density ratio $\frac{p}{q}$ on the gradient is scaled by the power of $(1-\alpha)$ as opposed to the unscaled ratio $\frac{p}{q}$ of the standard KL divergence (Regli & Silva, 2018). This ratio determines how much we can penalize the posterior with the prior. With the flexibility of choosing $\alpha$, the model can adjust the degree of prior penalization. Note that when $\alpha \in (0,1)$, $(\frac{p}{q})^{1-\alpha} < \frac{p}{q}$ when $\frac{p}{q} > 1$ and $(\frac{p}{q})^{1-\alpha} > \frac{p}{q}$ when $\frac{p}{q} < 1$, which means less penalty will be applied to the overestimated region of the prior where $\frac{p}{q} > 1$. When the prior is misspecified, choosing the RD can lead to a more robust posterior that focuses more on improving the likelihood and less on reducing the divergence (Futami et al., 2018; Regli & Silva, 2018).

## 3 Rényi Neural Processes

In this section we describe our Rényi Neural Process (RNP) framework, a simple yet effective strategy which provides a more robust way to learn neural processes without changing the model. We start by analyzing the main limitation of the standard neural process objective and present a motivating example. We illustrate a case where the prior is misspecified and describe our new objective with the RD to mitigate this.

### 3.1 Motivation: Standard neural processes and prior misspecification

We first consider a formal definition of prior misspecification.

**Definition 3.1.** *(Prior misspecification (Huang et al., 2024)) Let $\eta_\#\mathbb{Q}_\varphi$ be a pushforward of a probability measure $\mathbb{Q}_\varphi$ parameterized by $\varphi$ under the map $\eta: \mathcal{X} \times \mathcal{Y} \to \mathcal{Z}$. Then, $\{\eta_\#\mathbb{Q}_\varphi : \varphi \in \Phi\}$ defines a set of distributions on the space $\mathcal{Z}$ induced by the model, and $\eta_\#\mathbb{P}$ is the pushforward of the ground truth measure $\mathbb{P}$. The prior model is misspecified if $\forall\varphi \in \Phi, \eta_\#\mathbb{Q}_\varphi \neq \eta_\#\mathbb{P}$*

The definition suggests that when the prior model $\eta_\#\mathbb{Q}_\varphi$ is misspecified, there exists no optimal parameter $\varphi^*$ that can represent the true prior $\eta_\#\mathbb{P}$. This translate to NPs as the approximate prior model $q_\varphi(\mathbf{z}|X_C, Y_C)$ in Eq 3 can not recover the ground truth prior $p(\mathbf{z}|X_C, Y_C)$ for any parameterization of $\varphi$. We now show how this definition can assist us to analyze how a misspecified prior model can hinder neural processes learning.

**Proposition 3.2.** *Due to the prior approximation in Eq 3, if the prior model is misspecified, meaning that $\forall\varphi \in \Phi, \eta_\#\mathbb{Q}_\varphi(X_C, Y_C) \neq \eta_\#\mathbb{P}(X_C, Y_C)$, the resulting $\mathcal{L}_{VI}$ is no longer a valid bound for the marginal likelihood $p(Y_T|X_T, X_C, Y_C)$.*

Detailed proof can be found in Supp A.2. Proposition 3.2 challenges the common assumption $p(\mathbf{z}|X_C, Y_C) \approx q(\mathbf{z}|X_C, Y_C)$ that most NPs make. Such assumption, as we will show next, may result in a biased estimate of the posterior variance which hinders NP training.

**Illustrative example**: Fig 2 provides an example of a misspecified prior for neural processes. *The objective of this example is to compare how the vanilla NP and our proposed RNP behave when*

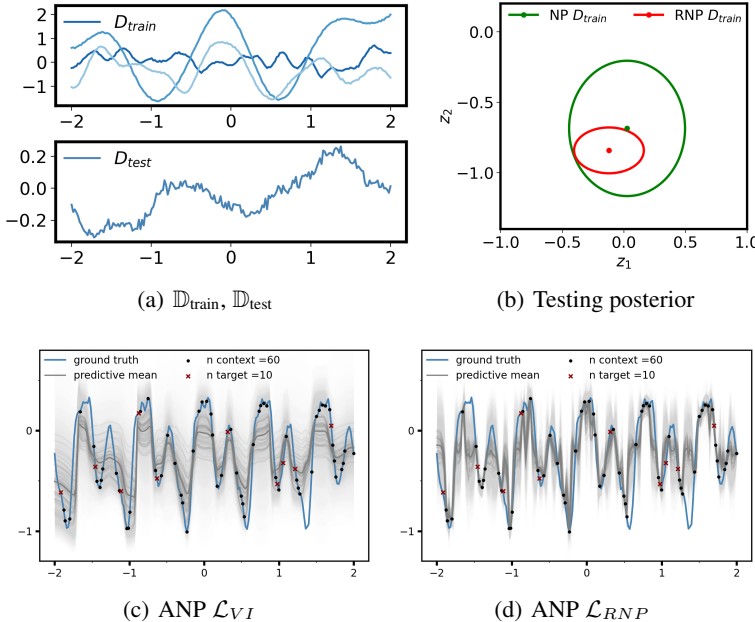

Figure 2: An illustrative example of prior misspecification. (a) $\mathbb{D}_{\text{train}} \sim \mathcal{GP}(\mathbf{0}, \mathcal{K}_{\text{RBF}})$, $\mathbb{D}_{\text{test}} \sim \mathcal{GP}(\mathbf{0}, \mathcal{K}_{\text{Matern}})$ are generated with different kernels to simulate prior misspecification. (b) Two Gaussian posteriors with $d_{\mathbf{z}} = 2$ conditioned on the context set $\mathbb{C}_{\text{test}}$ of $\mathbb{D}_{\text{test}} \sim \mathcal{GP}(\mathbf{0}, \mathcal{K}_{\text{Matern}})$: $q(\mathbf{z}|\mathbb{C}_{\text{test}}, \varphi_{NP_{D_{\text{train}}}})$ and $q(\mathbf{z}|\mathbb{C}_{\text{test}}, \varphi_{RNP_{D_{\text{train}}}})$. The RNP predicts lower variance estimates by restricting the impacts of consistency regularization caused by the prior misspecification to avoid oversmoothed predictions. (c) Predictive results with Attentive neural processes (ANP) (Kim et al., 2019) on $\mathbb{D}_{\text{test}} \sim \mathcal{GP}(\mathbf{0}, \mathcal{K}_{\text{periodic}})$ using the VI objective. The predictive mean underfits the data due to the consistency regularizer of prior misspecification in standard KL optimization. (d) ANP trained with our RNP objective dampens the consistency regularizer via the RD and explores the expressiveness of posterior to better fit the context points and provide better uncertainty estimate.

*the prior is misspecified.* We delay the introduction of the formulation of $\mathcal{L}_{RNP}$ in the next section, which is unnecessary for the illustration.

Let $\mathbb{D}_{\text{train}}$ and $\mathbb{D}_{\text{test}}$ denote datasets (meta-tasks) generated by two Gaussian Processes (GPs) with different kernels. We first train an NP model on $\mathbb{D}_{\text{train}}$ using the vanilla NP objective: $\varphi_{NP_{D_{\text{train}}}}, \theta^* = \min_{\varphi, \theta} \mathcal{L}_{VI}(\varphi, \theta; \mathbb{D}_{\text{train}})$. With that, we then fix the likelihood parameter $\theta^*$ in order to isolate the effect of the likelihood model on the predictive performance from the prior model and obtain another posterior model using the RNP objective: $\varphi_{RNP_{D_{\text{train}}}} = \min_{\varphi} \mathcal{L}_{RNP}(\varphi, \theta^*; \mathbb{D}_{\text{train}})$. Now consider evaluating both models using $\mathbb{D}_{\text{test}}$ by conditioning on these parameters and the test context set $\mathbb{C}_{\text{test}}$, then $q(\mathbf{z}|\mathbb{C}_{\text{test}}, \varphi_{NP_{D_{\text{train}}}})$ and $q(\mathbf{z}|\mathbb{C}_{\text{test}}, \varphi_{RNP_{D_{\text{train}}}})$ can be viewed as misspecified prior models since their parameters were optimized on $\mathbb{D}_{\text{train}}$ which looks quite different from $\mathbb{D}_{\text{test}}$. The resulting posteriors in Fig 2(b) show that RNPs obtaine a much smaller variance estimate, suggesting that the consistency regularizer $D_{\text{KL}}\left(q_{\varphi}(\mathbf{z}|\mathbb{T}, \mathbb{C})\|q_{\varphi}(\mathbf{z}|\mathbb{C})\right)$ is too strong in the vanilla NPs that it forces the posterior to produce a higher variance. As a result, vanilla NPs produce oversmoothed predictions that underfit the data as shown in Fig 2(c). In contrast, our RNP dampens the impacts of prior misspecification via the RD divergence and our predictive model respect the expressiveness of the posterior model better and produce superior predictive means as well as uncertainty estimate shown in Fig 2(d).

Hence, we argue that using RNP is beneficial when existing NP models have only limited capacity/data which could induce prior misspecification. We will now introduce our new neural process learning method, which can be generally applied to NPs using either VI or ML objectives.

## 3.2 PROPOSED METHOD: NEURAL PROCESSES WITH THE NEW OBJECTIVES

**New objective for VI-based NPs.** The main issue of NPs is the prior approximation $q_{\varphi}(\mathbf{z}|X_C, Y_C)$ wrt the true prior $p(\mathbf{z}|X_C, Y_C)$. In this case, the posterior variance may be critically overestimated

in some regions and underestimated in others. We therefore seek to obtain an alternative posterior distribution to alleviate this prior misspecification. Depending on whether the original NP framework is trained using the VI or the ML objective, we can revise the objective by minimizing the RD instead of KLD on two distributions. More specifically, in the case of VI-NPs where the inference of the latent variable $q(\mathbf{z})$ is of interest, the RD between the the approximated posterior distribution $q_\varphi(\mathbf{z}|X_T, Y_T, X_C, Y_C)$ and the true posterior $p(\mathbf{z}|X_T, Y_T, X_C, Y_C)$ is minimized.

$$\min_{\theta,\varphi} D_\alpha \left( q_\varphi(\mathbf{z}|X_T, Y_T, X_C, Y_C) \| p(\mathbf{z}|X_T, Y_T, X_C, Y_C) \right) \tag{6a}$$

$$\approx \max_{\theta,\varphi} \frac{1}{1-\alpha} \log \mathbb{E}_{q_\varphi(\mathbf{z}|X_T, Y_T, X_C, Y_C)} \left[ \frac{p_\theta(Y_T|X_T, \mathbf{z}) q_\varphi(\mathbf{z}|X_C, Y_C)}{q_\varphi(\mathbf{z}|X_T, Y_T, X_C, Y_C)} \right]^{1-\alpha} \tag{6b}$$

where details can be found in A.4. Eq 6b is an approximation obtained by replacing $p(\mathbf{z}|X_C, Y_C)$ with $q_\varphi(\mathbf{z}|X_C, Y_C)$. We can approximate the intractable expectation with Monte Carlo:

$$-\mathcal{L}_{RNP}(\theta, \varphi) = \frac{1}{1-\alpha} \mathbb{E}_{\mathbb{D}_{\text{train}}} \left[ \log \frac{1}{K} \sum_{k=1}^{K} \left[ \frac{p_\theta(Y_T|X_T, \mathbf{z}_k) q_\varphi(\mathbf{z}_k|X_C, Y_C)}{q_\varphi(\mathbf{z}_k|X_T, Y_T, X_C, Y_C)} \right]^{1-\alpha} \right], \quad \mathbf{z}_k \sim q_\varphi(\mathbf{z}|X_T, Y_T, X_C, Y_C). \tag{7}$$

**Prior, posterior and likelihood models.** One main advantage of RNP is we do not need to change the parameters of interests of the original NP models. Therefore, for the likelihood model $p_\theta(Y_T|X_T, \mathbf{z})$ we can adopt simple model architectures like NPs (Garnelo et al., 2018b) which assume independence between target points $p_\theta(Y_T|X_T, \mathbf{z}) = \prod_{n=1}^{N} p_\theta(\mathbf{y}_n|\mathbf{x}_n, \mathbf{z})$. The distribution of each target point is then modelled as Gaussian $p_\theta(\mathbf{y}_n|\mathbf{x}_n, \mathbf{z}) = \mathcal{N}(h_\mu(\mathbf{x}_n, \mathbf{z}), \text{Diag}(h_\sigma(\mathbf{x_n}, \mathbf{z})))$, and the decoder networks $h_\mu$ and $h_\sigma$ map the concatenation of the input feature $\mathbf{x}_n$ and $\mathbf{z}$ to the distribution parameters.

The prior model $q_\varphi(\mathbf{z}|X_C, Y_C)$ is more interesting as it is a set-conditional distribution and we are supposed to sample from it and evaluate the density of the samples. One feasible solution is to define a parametric distribution on a DeepSet (Zaheer et al., 2017). For instance, $q_\varphi(\mathbf{z}|X_C, Y_C) = \mathcal{N}(g_\mu(h(\mathbb{C})), \text{Diag}(g_\sigma(h(\mathbb{C}))))$ where $h(\mathbb{C}) = \frac{1}{|\mathbb{C}|} \sum_{m=1}^{|\mathbb{C}|} h(\mathbf{x}_m, \mathbf{y}_m)$ is a DeepSet function on the context set $\mathbb{C}$. In practice diagonal Gaussian distributions worked well with high dimensional latent variables $\mathbf{z}$. ANPs (Kim et al., 2019) incorporate dependencies between context points $q_\varphi(\mathbf{z}|\mathbb{C}) = q_\varphi(\mathbf{z}|\mathbf{x}_{1:m}, \mathbf{y}_{1:m})$ using self-attention networks. But one can consider more flexible distributions such as conditional normalising flows (Luo et al., 2023) for sample and density estimation. As previously stated, the posterior distribution is defined by coupling its parameters with the prior. Therefore the posterior $q_\varphi(\mathbf{z}|X_T, Y_T, X_C, Y_C)$ in the DeepSet case can be represented as $q_\varphi(\mathbf{z}|X_T, Y_T, X_C, Y_C) = \mathcal{N}(g_\mu(h(\mathbb{C}, \mathbb{T})), \text{Diag}(g_\sigma(h(\mathbb{C}, \mathbb{T}))))$. To apply stochastic gradient descent over the parameters of the posterior, we applied the reparameterization trick to obtain samples $\mathbf{z}_k = g_\sigma^{\frac{1}{2}}(h(\mathbb{C}, \mathbb{T})) * \epsilon + g_\mu(h(\mathbb{C}, \mathbb{T})), \epsilon \sim \mathcal{N}(\mathbf{0}, \mathbf{I})$.

**New objective for ML-based NPs.** As the goal of NPs is to maximize predictive likelihood instead of inferring the latent distribution, another type of NPs directly parameterize the likelihood model without explicitly defining the latent variable $\mathbf{z}$. Following Futami et al. (2018), we can rewrite the maximum likelihood estimation as minimizing the KLD between the empirical distribution $\hat{p}(\mathbf{y}|\mathbf{x}, \mathbb{C})$ and the model distribution $p(\mathbf{y}|\mathbf{x}, \mathbb{C}, \theta)$:

$$-\mathcal{L}_{ML}(\theta) = \max_\theta \mathbb{E}_{\mathbb{D}_{\text{train}}} \log p_\theta(Y_T|X_T, \mathbb{C}) \tag{8}$$

$$\equiv \max_\theta \mathbb{E}_{\mathbb{D}_{\text{train}}} \left[ \frac{1}{N} \sum_{n=1}^{N} \log p_\theta(\mathbf{y}_n|\mathbf{x}_n, \mathbb{C}) \right] \approx \min_\theta \mathbb{E}_{\mathbb{D}_{\text{train}}} \left[ D_{\text{KL}}(\hat{p}(\mathbf{y}|\mathbf{x}, \mathbb{C}) \| p_\theta(\mathbf{y}|\mathbf{x}, \mathbb{C})) \right] \tag{9}$$

where $\hat{p}(\mathbf{y}|\mathbf{x}, \mathbb{C})$ is the empirical distribution defined as $\frac{1}{N} \sum_{n=1}^{N} \delta(\mathbf{y}, \mathbf{y}_n)$ where $y_n$ are samples from the unknown distribution $p^*(\mathbf{y}|\mathbf{x}, \mathbb{C})$. Replacing the KLD with RD we get:

$$\min_\theta \mathbb{E}_{\mathbb{D}_{\text{train}}} D_\alpha (\hat{p}(\mathbf{y}|\mathbf{x}, \mathbb{C}) \| p_\theta(\mathbf{y}|\mathbf{x}, \mathbb{C})) \approx \min_\theta \mathbb{E}_{\mathbb{D}_{\text{train}}} \frac{1}{N} \sum_{n=1}^{N} \frac{1}{\alpha - 1} \log p_\theta^{1-\alpha}(\mathbf{y}_n|\mathbf{x}_n, \mathbb{C}) + Const \tag{10}$$

$$\mathcal{L}_{RNPML}(\theta) = \mathbb{E}_{\mathbb{D}_{\text{train}}} \frac{1}{(\alpha-1)N} \sum_{n=1}^{N} \log p_\theta^{1-\alpha}(\mathbf{y}_n|\mathbf{x}_n, \mathbb{C}) = \frac{1}{(\alpha-1)N} \sum_{n=1}^{N} \log \left( \int p_\theta(\mathbf{y}_n, \mathbf{z}|\mathbf{x}_n, \mathbb{C}) d\mathbf{z} \right)^{1-\alpha} \tag{11}$$

where details can be found in A.5. Note $\alpha = 0$ corresponds to maximizing likelihood estimation and the new RNP objective essentially reweights samples based on their likelihood. We define $p_\theta(\mathbf{y}_n, \mathbf{z}|\mathbf{x}_n, \mathbb{C}) = p_\theta(\mathbf{y}_n, \mathbf{z}|\mathbf{x}_n)p(\mathbf{z}|\mathbb{C})$ and use $p_\varphi(\mathbf{z}|X_C, Y_C) \approx p(\mathbf{z}|\mathbb{C})$ . Then Eq 10 can be approximated with Monte Carlo:

$$\mathcal{L}_{RNPML}(\theta, \varphi) \approx \mathbb{E}_{\mathbb{D}_{\text{train}}} \frac{1}{(\alpha-1)N} \sum_{n=1}^{N} \log \left( \frac{1}{K} \sum_{k=1}^{K} p_\theta(\mathbf{y}_n|\mathbf{z}_k, \mathbf{x}_n) \right)^{1-\alpha}, \quad \mathbf{z}_k \sim p_\varphi(\mathbf{z}|X_C, Y_C) \tag{12}$$

One advantage of ML-based method is that we do not need to estimate the density of the samples from the prior model. Therefore, reparameterization tricks (Kingma et al., 2015) can be applied to obtain samples from flexible prior models: $\mathbf{z}_k = s_\varphi(X_C, Y_C, \epsilon_k), \epsilon_k \sim \mathcal{N}(\mathbf{0}, I)$ where $s_\varphi$ is a neural network.

## 4 PROPERTIES OF RNPS

We generalize the theorem from Rényi variational inference (Li & Turner, 2016) to RNPs:

**Theorem 4.1.** *(Monotonicity (Li & Turner, 2016))*

$\mathcal{L}_{RNP}$ *is continuous and non-increasing with respect to the hyper-parameter $\alpha$.*

**Proposition 4.2.** *(Unification of the objectives)*

$$\mathcal{L}_{ML} = \mathcal{L}_{RNP,\alpha=0} \geq \mathcal{L}_{RNP,\alpha\in(0,1)} \geq \mathcal{L}_{RNP,\alpha\to 1} = \mathcal{L}_{VI}.$$

Theorem 4.1 and Proposition 4.2 (Proof see Supp A.6) suggest that these objectives are bounded by $\log \int p(Y_T|X_T, \mathbf{z})q(\mathbf{z}|X_C, Y_C)$. However, it is not the marginal likelihood due to the approximation $q(\mathbf{z}|X_C, Y_C) \approx p(\mathbf{z}|X_C, Y_C)$, and the training of NPs using these objectives does not guarantee the validity of the approximation. Therefore, the optimal $\alpha$ that achieves the best likelihood might not always correspond to 0, i.e., the maximum likelihood objective does not always guarantees the best marginal likelihood, and we can still improve the test likelihood using our RNP objective. We show in our experiment that benefits can be readily obtained by tuning $\alpha$ on a validation set.

**Tuning $\alpha \in (0, 1)$ can mitigate prior misspecification in RNPs**. We have shown that prior misspecification can be handled by improving the mass-covering ability in NPs. We will present how RNP can achieve this objective through the gradients of the parameters of the encoder networks $\varphi$ (more details can be found in Supp A.7):

$$\nabla_\varphi \mathcal{L}_{RNP} = \mathbb{E}_{q_\varphi(\mathbf{z}|\mathbb{C},\mathbb{T})} \left[ w_\alpha(\mathbf{z}, X_T, Y_T, \mathbb{C}) \nabla_\varphi \log \frac{p_\theta(Y_T|X_T, \mathbf{z})q_\varphi(\mathbf{z}|\mathbb{C})}{q_\varphi(\mathbf{z}|\mathbb{C}, \mathbb{T})} \right] \tag{13a}$$

$$\text{with } w_\alpha(\mathbf{z}, X_T, Y_T, \mathbb{C}) = \left( \frac{p_\theta(Y_T|X_T, \mathbf{z})q_\varphi(\mathbf{z}|\mathbb{C})}{q_\varphi(\mathbf{z}|\mathbb{C}, \mathbb{T})} \right)^{1-\alpha} \Bigg/ \mathbb{E}_{q(\mathbf{z}|\mathbb{C},\mathbb{T})} \left[ \frac{p_\theta(Y_T|X_T, \mathbf{z})q_\varphi(\mathbf{z}|\mathbb{C})}{q_\varphi(\mathbf{z}|\mathbb{C}, \mathbb{T})} \right]^{1-\alpha} \tag{13b}$$

The influence of the density ratio $\frac{q(\mathbf{z}|\mathbb{T},\mathbb{C})}{q(\mathbf{z}|\mathbb{C})}$ on the gradients is now scaled by $[\frac{q(\mathbf{z}|\mathbb{C})}{q(\mathbf{z}|\mathbb{T},\mathbb{C})}]^{1-\alpha}$ and $[\frac{q(\mathbf{z}|\mathbb{T},\mathbb{C})}{q(\mathbf{z}|\mathbb{C})}]^\alpha$ respectively. Tuning $\alpha \in (0, 1)$ offers the flexibility of how much we intend to penalize the gradients on the different regions of the posterior $q(\mathbf{z}|\mathbb{T}, \mathbb{C})$ and $q(\mathbf{z}|\mathbb{C})$.

## 5 INFERENCE WITH RÉNYI NEURAL PROCESSES

During inference time, as we cannot access the ground truth for the target outputs $Y_T$, we use the approximate prior $q(\mathbf{z}|X_C, Y_C)$ instead of the posterior distribution $q(\mathbf{z}|X_T, Y_T, X_C, Y_C)$ to estimate the marginal distribution:

$$p(Y_T|X_T, X_C, Y_C) = \int p_\theta(Y_T|X_T, \mathbf{z})q_\varphi(\mathbf{z}|X_C, Y_C)d\mathbf{z} \approx \frac{1}{K} \sum_{k=1}^{K} p_\theta(Y_T|X_T, \mathbf{z}_k), \mathbf{z}_k \sim q_\varphi(\mathbf{z}|X_C, Y_C). \tag{14}$$

We now provide the pseudo code for Rényi Neural Processes in Supp Algorithm 1. In addition to vanilla neural processes, our framework can also be generalized to other neural process variants as shown in the experiments section.

## 6 RELATED WORK

**Neural processes family.** Neural processes (Garnelo et al., 2018b) and conditional neural processes (Garnelo et al., 2018a) were initially proposed for the meta learning scheme where they make predictions given a few observations as context. Both of them use deepset models (Zaheer et al., 2017) to map a finite number of data points to a high dimensional vector and their likelihood models assume independencies among data points. The main difference is whether estimating likelihood maximization directly or introducing the latent variable and adopting variational inference framework. Under the existing NP setting, more members were introduced with different inductive biases in the model (Jha et al., 2022; Bruinsma et al., 2023; Dutordoir et al., 2023; Jung et al., 2024; Vadeboncoeur et al., 2023). For instance, attentive neural processes (Kim et al., 2019) incorporated dependencies between observations with attention neural works. Convolutional neural processes (Foong et al., 2020; Huang et al., 2023) assume translation equivariance among data points. These two methods explicitly defined the latent variable which requires density estimation. Recent works such as transformer neural processes (Nguyen & Grover, 2022) and neural diffusion processes (Dutordoir et al., 2023) turn to marginal likelihood maximization and do not have the latent distribution. More neural processes which claim to provide exact (Markou et al., 2022) or tractable inference (Lee et al., 2023; Wang et al., 2023) are introduced. Stable neural processes (Liu et al., 2024) argued that NPs are prone to noisy context points and proposed a weighted likelihood model that focuses on subsets that are difficult to predict, but do not focus regularizing the posterior distribution. Compared to variational inference based methods, non-VI predictions can be less robust to noisy inputs in the data (Futami et al., 2018). It is also challenging to incorporate prior knowledge (Zhang et al., 2018) into these neural processes, which could be beneficial when no data is observed for the task.

**Robust divergences.** Divergences in variational inference can be viewed as an regularization on the posterior distribution via the prior distribution. The commonly adopted KL divergence which minimizes the expected density ratio between the posterior is infamous for underestimating the true variance of the target distribution (Regli & Silva, 2018). Several other divergences have since been proposed to focus on obtaining a robust posterior when the input and output features are noisy or the existence of outliers. Examples of robust divergences include Rényi divergence (Lee & Shin, 2022), beta and alpha divergences (Futami et al., 2018; Regli & Silva, 2018) and f-divergences (Cheng et al., 2021). They provide the extra parameters or flexible functions to control the density ratio so that the posterior can focus more on mass covering, mode seeking abilities or is robust against outliers. Generalized variational inference (Knoblauch et al., 2019) suggested that any form of divergence can be used to replace the KL objective when the model is misspecified. As neural processes facilitate a posterior to approximate the prior distribution, we are able to control the density ratios of two posterior distributions with robust divergences for better predictions.

## 7 EXPERIMENTS

**Datasets and training details.** We evaluate the proposed method on multiple regression tasks: 1D regression (Garnelo et al., 2018a; Gordon et al., 2019; Kim et al., 2019; Nguyen & Grover, 2022), image inpainting (Gordon et al., 2019; Nguyen & Grover, 2022). 1D regression includes three Gaussian Process (GP) regression tasks with different kernels: RBF, Matern 5/2 and Periodic. Given a function $f$ sampled from a GP prior with varying scale and length and a context set generated by such function, our goal is to predict the target distribution. The number of context points is randomly sampled $M \sim \mathcal{U}(3, 50)$, and the number of target points is $N \sim \mathcal{U}(3, 50 - M)$ (Nguyen & Grover, 2022). We choose 100,000 functions for training, and sample another 3,000 functions for testing. The input features were normalized to $[-2, 2]$. Image inpainting involves 2D regression on three image datasets: MNIST, SVHN and CelebA. Given some pixel coordinates $\mathbf{x}$ and intensities $\mathbf{y}$ as context, the goal is to predict the pixel value for the rest of image. The number of context points for inpainting tasks is $M \sim \mathcal{U}(3, 200)$ and the target point count is $N \sim \mathcal{U}(3, 200 - M)$. The input coordinates were normalized to $[-1, 1]$ and pixel intensities were rescaled to $[-0.5, 0.5]$. All the models can be trained using a single GPU with 16GB memory.

**Baselines.** We first validate our approach on state-of-the-art NP families: neural processes (NP) (Garnelo et al., 2018b), attentive neural processes (ANP) (Kim et al., 2019), Bayesian aggregation neural

processes (BA-NP) (Volpp et al., 2021)[1], transformer neural processes with diagonal covariances (TNP-D) (Nguyen & Grover, 2022)[2], and versatile neural processes (VNP) (Guo et al., 2023)[3]. For VNPs, they chose different parameterizations for the prior and posterior models, which can be used to validate if our objective is superior than simply decoupling the two models. We generalize the NP objective to RNP using Eq 12 or Eq 7 depending on whether the baseline model infer the latent distribution $p(\mathbf{z})$. The methods are considered as a special case of $\alpha = 1$ of RNP if the baseline model uses the VI objective or $\alpha = 0$ if the baseline model uses the ML-objective. The number of samples $K$ for the Monte Carlo is 32 for training and 50 for inference. Our experiments aim to answer the following research questions: (1) Can RNPs achieve better performance over existing NP frameworks? (2) How does the model perform under prior misspecifications? (3) How to select the optimal $\alpha$ values? We also carry out ablation studies investigating how to select the optimal $\alpha$ values, the number of MC samples and the number of context points for our RNP framework.

## 7.1 Predictive performance

We compare the test log-likelihood on both the context and target sets across different datasets. Specifically, we adopted the VI-based RNP objective to train NPs, ANPs and VNPs as their model designs include the prior models. We used the ML-based RNP objective to train TNP and BANP because the TNP objective was originally defined using ML only and the ML objective significantly outperformed the VI objective for BANPs. We set $\alpha = 0.7$ to train for VI-based RNPs and analogously $\alpha = 0.3$ for ML-based baselines. However, we show in section 7.3 that, in fact, these values can be set optimally via cross-validation. To put the baseline models in the spectrum, $\alpha = 1$ corresponds to the standard VI solutions (using the KLD), and $\alpha = 0$ corresponds to the maximum likelihood solutions.

Table 1 shows the mean test log-likelihood $\pm$ one standard deviation using 5 different random seeds for each method. We see that RNP consistently improved log-likelihood over the other two objectives and ranked the highest for all the baselines. RNP also consistently achieved better likelihood on TNP-D and VNP which generally outperform other baseline models across datasets. Some prominent improvements were achieved in harder tasks in 1D regression such as ANP Periodic and BA-NP Periodic where the vanilla NP objectives underperform. As previously illustrated in Fig 2(c) and Fig 2(d), RNP improves predictive performance by mitigating the oversmoothed predictions on periodic data. This could suggest that a misspecified prior model in the vanilla NP objective imposes an unjustifiable regularization on the posterior and hinder the expressiveness of the posterior and consequently predictive performance. RNP also significantly improved test likelihood of BA-NP and VNP on image inpainting tasks, demonstrating the superiority of RNPs on higher dimensional data.

## 7.2 Prior misspecification

In order to isolate the impacts of data and paramertization on the prior models $q(\mathbf{z}|\mathbb{C}, \varphi)$, we designed two sets of experiments for validation: $q(\mathbf{z}|\mathbb{C}_{bad}, \varphi)$ which is conditioned on poor context data and $q(\mathbf{z}|\mathbb{C}, \varphi_{bad})$ where the paramerization does not fit with the clean context data (Section 7.1 belongs to this category). We corrupt the context data with random noise for the former and domain shift datasets for the latter scenario (more detailed settings can found in section A.8). Table 2 shows the test log-likelihood for the high-performing baseline TNP-D on two misspecified cases where tasks are generated from different distributions during the meta training and meta testing phase and Supp Table 4 shows test log-likelihood under noisy context. For the 1D regression task, the model is trained using the Lotka-Volterra dataset which is generally used for prey-predator simulations. The dynamics is controlled by a two-variable ordinary differential equations: $\dot{x} = \theta_1 x - \theta_2 xy, \dot{y} = -\theta_3 y + \theta_4 xy$ where $x$ and $y$ correspond to the populations of the prey and predator respectively. The parameters are chosen as $\theta_1 = 1, \theta_2 = 0.01, \theta_3 = 0.5, \theta_4 = 0.01$ following (Gordon et al., 2019). The number of context points is randomly sampled $M \sim \mathcal{U}(15, 100)$, and the number of target points is $N \sim \mathcal{U}(15, 100 - M)$. We choose 20,000 functions for training, and sample another 1,000 functions for evaluation. We then test the model on a real-world Hare-Lynx dataset which tracks the two species populations over 90 years. The input and output features were normalized via z-score

---

[1]https://github.com/boschresearch/bayesian-context-aggregation

[2]https://github.com/tung-nd/TNP-pytorch/tree/master/regression

[3]https://github.com/ZongyuGuo/Versatile-NP

Table 1: Test set log-likelihood ↑. The best performance results for each dataset are in **bold**.

| Model | Set | Objective | RBF | Matern 5/2 | Periodic | MNIST | SVHN | CelebA | Avg rank |
|---|---|---|---|---|---|---|---|---|---|
| NP (Garnelo et al., 2018b) | context | $\mathcal{L}_{VI}$ | 0.69±0.01 | 0.56±0.02 | -0.49±0.01 | 0.99±0.01 | 3.24±0.02 | 1.71±0.04 | 2.3 |
| | | $\mathcal{L}_{ML}$ | 0.68±0.02 | 0.55±0.02 | **-0.48±0.03** | 1.00±0.01 | 3.22±0.03 | 1.70±0.03 | 2.5 |
| | | $\mathcal{L}_{RNP(\alpha)}$ | **0.78±0.01** | **0.66±0.01** | -0.49±0.00 | **1.01±0.02** | **3.26±0.01** | **1.72±0.05** | **1.2** |
| | target | $\mathcal{L}_{VI}$ | 0.26±0.01 | 0.09±0.02 | **-0.61±0.00** | 0.90±0.01 | 3.08±0.01 | 1.45±0.03 | 2.3 |
| | | $\mathcal{L}_{ML}$ | 0.28±0.02 | 0.11±0.02 | -0.61±0.01 | **0.92±0.01** | 3.07±0.02 | **1.47±0.02** | 1.8 |
| | | $\mathcal{L}_{RNP(\alpha)}$ | **0.33±0.01** | **0.16±0.01** | -0.62±0.00 | 0.91±0.01 | **3.09±0.01** | 1.45±0.03 | **1.7** |
| ANP (Kim et al., 2019) | context | $\mathcal{L}_{VI}$ | **1.38±0.00** | **1.38±0.00** | 0.65±0.04 | **1.38±0.00** | 4.14±0.00 | 3.92±0.07 | 1.3 |
| | | $\mathcal{L}_{ML}$ | **1.38±0.00** | **1.38±0.00** | 0.63±0.03 | **1.38±0.00** | **4.14±0.01** | 3.86±0.07 | 1.7 |
| | | $\mathcal{L}_{RNP(\alpha)}$ | **1.38±0.00** | **1.38±0.00** | **1.22±0.02** | **1.38±0.00** | **4.14±0.00** | **3.97±0.03** | **1.0** |
| | target | $\mathcal{L}_{VI}$ | 0.81±0.00 | 0.64±0.00 | -0.91±0.02 | **1.06±0.01** | **3.65±0.01** | **2.24±0.03** | 1.7 |
| | | $\mathcal{L}_{ML}$ | 0.80±0.00 | 0.64±0.00 | -0.89±0.02 | 1.04±0.01 | **3.65±0.01** | 2.23±0.03 | 2.2 |
| | | $\mathcal{L}_{RNP(\alpha)}$ | **0.84±0.00** | **0.67±0.00** | **-0.57±0.01** | 1.05±0.01 | 3.61±0.02 | **2.24±0.02** | **1.3** |
| BA-NP (Volpp et al., 2021) | context | $\mathcal{L}_{VI}$ | 1.43±0.03 | **1.04±0.08** | -0.65±0.84 | 0.81±0.84 | 2.76±0.59 | 1.65±0.01 | 2.3 |
| | | $\mathcal{L}_{ML}$ | 1.30±0.09 | 0.69±0.05 | -0.70±0.07 | 3.62±0.06 | 4.87±0.05 | **2.02±0.02** | 2.3 |
| | | $\mathcal{L}_{RNP(\alpha)}$ | **1.45±0.04** | 1.01±0.04 | **-0.41±0.02** | **3.85±0.09** | **4.88±0.05** | 2.00±0.02 | **1.3** |
| | target | $\mathcal{L}_{VI}$ | 1.19±0.03 | **0.79±0.09** | -0.89±0.01 | 0.24±0.64 | 2.60±0.50 | 1.30±0.01 | 2.5 |
| | | $\mathcal{L}_{ML}$ | 1.12±0.08 | 0.53±0.04 | -0.91±0.05 | 3.56±0.06 | 4.29±0.04 | **1.63±0.01** | 2.3 |
| | | $\mathcal{L}_{RNP(\alpha)}$ | **1.22±0.04** | **0.79±0.03** | **-0.72±0.02** | **3.79±0.09** | **4.31±0.02** | 1.61±0.02 | **1.2** |
| TNP-D (Nguyen & Grover, 2022) | context | $\mathcal{L}_{ML}$ | 2.58±0.01 | 2.57±0.01 | **-0.52±0.00** | 1.73±0.11 | 10.63±0.12 | 4.61±0.27 | 1.8 |
| | | $\mathcal{L}_{RNP(\alpha)}$ | **2.59±0.01** | **2.59±0.00** | **-0.52±0.00** | **1.81±0.12** | **10.72±0.08** | **4.66±0.23** | **1.0** |
| | target | $\mathcal{L}_{ML}$ | 1.38±0.01 | 1.03±0.00 | **-0.59±0.00** | 1.63±0.07 | 6.69±0.04 | 2.45±0.05 | 1.8 |
| | | $\mathcal{L}_{RNP(\alpha)}$ | **1.41±0.00** | **1.04±0.00** | **-0.59±0.00** | **1.67±0.07** | **6.71±0.04** | **2.46±0.06** | **1.0** |
| VNP (Guo et al., 2023) | context | $\mathcal{L}_{VI}$ | 1.37±0.00 | 1.37±0.00 | 1.23±0.03 | 1.60±0.10 | 0.80±0.00 | 0.08±0.03 | 2.0 |
| | | $\mathcal{L}_{RNP(\alpha)}$ | **1.38±0.00** | **1.38±0.00** | **1.32±0.01** | **3.63±0.39** | **4.00±0.06** | **2.65±0.06** | **1.0** |
| | target | $\mathcal{L}_{VI}$ | 0.90±0.02 | 0.70±0.03 | -0.49±0.00 | 1.59±0.10 | 0.80±0.00 | 0.08±0.03 | 2.0 |
| | | $\mathcal{L}_{RNP(\alpha)}$ | **0.92±0.01** | **0.71±0.03** | **-0.48±0.00** | **3.62±0.37** | **3.89±0.06** | **2.49±0.06** | **1.0** |

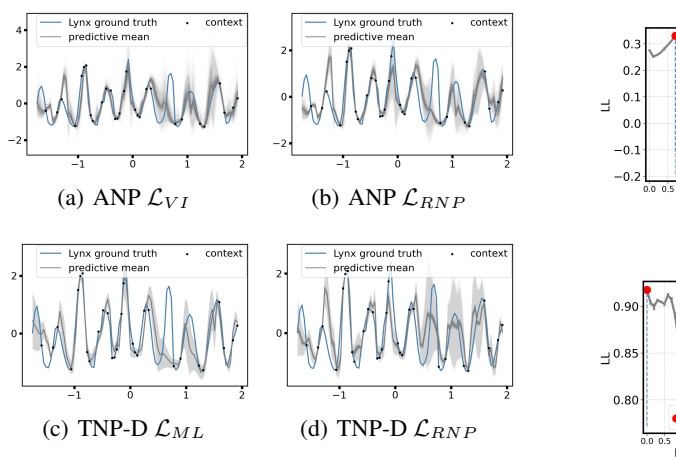

(a) ANP $\mathcal{L}_{VI}$    (b) ANP $\mathcal{L}_{RNP}$

(c) TNP-D $\mathcal{L}_{ML}$    (d) TNP-D $\mathcal{L}_{RNP}$

Figure 3: Prior misspecification experiment. Both models are trained on simulated Lotka-Volterra data and tested on the real-world Hare-Lynx dataset.

(a) $\alpha$ on RBF

(b) $\alpha$ on MNIST

Figure 4: Hyperparameter($\alpha$) tuning.

normalization. Our method in table 4 shows outperformance across multiple datasets as the impact of misspecified contexts is alleviated via the divergence. The results in table 2 show that RNP significantly outperformed the ML objective on both the training and testing data, highlighting the robustness of our objective. Fig 3 shows the prediction results on the Lynx dataset, where the RNP achieves better uncertainty estimate and tracks the seasonality of the data more efficiently than the ML objective. We also tested TND-D on the Extended MNIST dataset with 47 classes that include letters and digits. We use classes 0-10 for meta training and hold out classes 11-46 for meta testing under prior misspecification. Table 2 shows that RNP performed slightly worse on the EMNIST training task but significantly outperformed the ML objective on the test set (last column), which demonstrates the superior robustness of the new objective under misspecification.

## 7.3 HYPER-PARAMETER TUNING

**How to select the optimal $\alpha$ values?** We have already shown choosing some $\alpha$ can obtain better log-likelihood across different datasets. We now demonstrate that one can find the optimal task-specific $\alpha$ value to further improve the performance. Following (Futami et al., 2018), we use cross

Table 2: Loglikelihood ↑ under prior misspecification using TNP-D.

| Objective | $\mathbb{D}_{\text{train}}$ (Lotka-Volterra) | | Misspec $\mathbb{D}_{\text{test}}$ (Hare-Lynx) | | $\mathbb{D}_{\text{train}}$ EMNIST (class 0-10) | | Misspec $\mathbb{D}_{\text{test}}$ (class 11-46) | |
|---|---|---|---|---|---|---|---|---|
| | context | target | context | target | context | target | context | target |
| $\mathcal{L}_{ML}$ | 3.09±0.22 | 1.98±0.11 | -0.59±0.47 | -4.44±0.41 | **1.54±0.05** | **1.56±0.07** | 0.03±0.97 | -0.20±0.57 |
| $\mathcal{L}_{RNP}$ | **3.32±0.15** | **2.12±0.06** | **-0.17±0.31** | **-3.63±0.09** | 1.52±0.08 | 1.47±0.12 | **0.96±0.18** | **0.70±0.15** |

validation for $\alpha$ selection. We hyper-searched the $\alpha$ values from 0 to 2 with an interval of 0.1. For $\alpha = 1$, we use the vanilla NP objective. As shown in Fig 4(a) and Fig 4(b), the optimal solutions are data-specific and model specific, but $\alpha \in (0, 1)$ generally improves the NP objective more than $\alpha > 1$. Empirically, we found $\alpha = 0.7$ for the VI objectives and $\alpha = 0.3$ for the ML objectives keep a good balance between performance improvement and prior regularization. We also provide a heuristic strategy to effecienctly train the RNP framework by annealing $\alpha$ from close to 1 and decreasing it to 0 (details can be found A.9).

## 7.4 ABLATION STUDIES

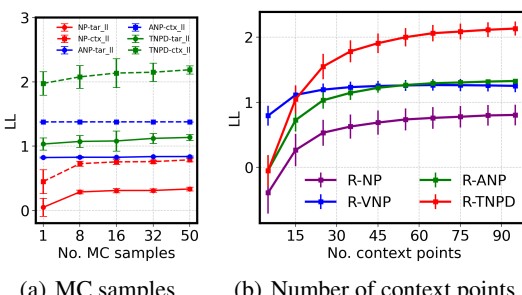

(a) MC samples          (b) Number of context points

Figure 5: Ablation study.

**Effects of Monte Carlo samples on likelihood.** As both RNP (Eq 7) and RNP-ML (Eq 12) require MC approximations, we investigate the effects of the number of MC samples $K$ on predictive likelihood. We set $K \in \{1, 8, 16, 32, 50\}$ for optimizing the RNP objective during training and use $K = 50$ for inference. Note that $K = 1$ corresponds to the deterministic NPs (conditional NPs). Fig 5(a) shows both the context and target log-likelihood for three methods: NP, ANP and TNP-D on the RBF dataset. As expected, increasing the number of MC samples improves the LL mean and also reduces the variance for all the methods with $K = 50$ achieving the highest LL and the smallest variance. In practice, we set $K = 32$ to balance performance and memory efficiency.

**Effects of the number of context points on likelihood.** We study the effect of the context points on the target likelihood. During training the number of context points is sampled from $\mathcal{U}(3, 50)$ and we vary the number of context points from 5 to 95 at an interval of 10 for evaluation. The results of RBF in Fig 5(b) shows that increasing the context set size leads to improved LL for all the methods. Most methods (e.g., NP, ABP, VNP) plateaued after the number of contexts increases to more than 45, whereas TNP-D still shows unsaturated performance improvement with the increased context size.

## 8 CONCLUSION

In this paper, we propose the Rényi Neural Process (RNP), a new NP framework designed to mitigate prior misspecification in neural processes. We bridge the commonly adopted variational inference and maximum likelihood estimation objectives in vanilla NPs through the use of Rényi divergence and demonstrate the superiority of our generalized objective in improving predictive performance by selecting optimal $\alpha$ values. We apply the framework to multiple state-of-the-art NP models and observe consistent log-likelihood improvements across benchmarks, including 1D regression, image inpainting, and real-world regression tasks. Our framework can be further extended to improve the robustness of contextual inferences, such as prompt design in large language models. Limitations of our framework lie in drawing multiple samples with Monte-Carlo which scarifies the computational efficiency in exchange for better performance and infeasibility to compute analytical solutions on the divergence.

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

# A  APPENDIX

## A.1  PSEUDOCODE

---

**Algorithm 1** Rényi Neural Processes

---
**Input**: Context inputs $X_C$ and outputs $Y_C$. Target inputs $X_T$, and target outputs $Y_T$ during training
**Output**: Target distribution $p(Y_T|X_T, X_C, Y_C)$
**Training:**
**for** epoch=1 **to** max_epoch **do**
  Sample a context set $(X_C, Y_C)$ and a target set $(X_T, Y_T)$
  Obtain the posterior distribution with the encoding network: $q_\varphi(\mathbf{z}|X_T, Y_T, X_C, Y_C)$
  Obtain the approximated prior distribution with the encoding network: $q_\varphi(\mathbf{z}|X_C, Y_C)$
  Sample $\mathbf{z}_1, ..., \mathbf{z}_K \sim q_\varphi(\mathbf{z}|X_T, Y_T, X_C, Y_C)$
  Construct the likelihood model with the decoding network: $p_\theta(Y_T|X_T, \mathbf{z}_k)$
  Compute the objective $\mathcal{L}_{RNP}$ using Eq 7 or Eq 12
  Update the encoder parameters with $\nabla_\varphi \mathcal{L}$ (Eq 13b) and the decoder parameters with $\nabla_\theta \mathcal{L}$
**end for**
**Inference:**
Construct the approximated prior $q_\varphi(\mathbf{z}|X_C, Y_C)$
Sample $\mathbf{z}_1, ..., \mathbf{z}_K \sim q_\varphi(\mathbf{z}|X_C, Y_C)$
Predict the target distribution $p_\theta(Y_T|X_T, \mathbf{z}_k)$
Estimate the log-likelihood of the target outputs using Eq 14.

---

Table 3: Notation

| Name | Description |
|------|-------------|
| $f$ | function sample from a stochastic process |
| $\mathcal{X}$ | input space |
| $\mathcal{Y}$ | output space |
| $\mathcal{Z}$ | latent space |
| $\mathbf{x}$ | input features |
| $\mathbf{y}$ | output features |
| $\mathbf{z}$ | latent variable representing stochasticity of the functional sample $f$ |
| $X_C$ | $\mathbb{R}^{M \times D}$ context inputs |
| $Y_C$ | $\mathbb{R}^{M \times 1}$ context outputs |
| $X_T$ | $\mathbb{R}^{N \times D}$ target inputs |
| $Y_T$ | $\mathbb{R}^{N \times 1}$ target outputs |
| $M$ | number of samples in the context set, indexed by $m$ |
| $N$ | number of samples in the target set, indexed by $n$ |
| $D_x$ | dimension of input features |
| $D_y$ | dimension of input features |
| $\mathbb{C}, \mathbb{T}$ | notation for Context and Target |
| $\varphi$ | parameters in the posterior model $q(\mathbf{z}|X, Y)$ |
| $\theta$ | parameters in the likelihood model $p(Y|\mathbf{z}, X)$ |
| $\eta$ | pushward of a measure $\mathbb{P} : \mathcal{X} \times \mathcal{Y} \to \mathcal{Z}$ |
| $K$ | number of $\mathbf{z}$ samples for Monte Carlo approximation |

## A.2 PROOF OF PROPOSITION 3.2

The true ELBO of neural processes without prior approximation can be written as:

$$ELBO = \mathbb{E}_{q_\varphi(\mathbf{z}|X_T,Y_T,X_C,Y_C)} \log p_\theta(Y_T|X_T,\mathbf{z}) - D_{\text{KL}}\left(q_\varphi(\mathbf{z}|X_T,Y_T,X_C,Y_C)\|p(\mathbf{z}|X_C,Y_C)\right)$$

$$\tag{15a}$$

$$= \mathbb{E}_{q_\varphi(\mathbf{z}|\mathbb{C},\mathbb{T})}[\log p_\theta(Y_T|X_T,\mathbf{z}) + \log p(\mathbf{z}|\mathbb{C}) - \log q_\varphi(\mathbf{z}|\mathbb{C},\mathbb{T}) + \log q_\varphi(\mathbf{z}|\mathbb{C}) - \log q_\varphi(\mathbf{z}|\mathbb{C})]$$

$$\tag{15b}$$

$$= \mathbb{E}_{q_\varphi(\mathbf{z}|\mathbb{C},\mathbb{T})}\left[\log \frac{p_\theta(Y_T|X_T,\mathbf{z})q_\varphi(\mathbf{z}|\mathbb{C})}{q_\varphi(\mathbf{z}|\mathbb{C},\mathbb{T})} + \log \frac{p(\mathbf{z}|\mathbb{C})}{q_\varphi(\mathbf{z}|\mathbb{C})}\right] \tag{15c}$$

$$= \mathbb{E}_{q_\varphi(\mathbf{z}|\mathbb{C},\mathbb{T})}\left[\log \frac{p_\theta(Y_T|X_T,\mathbf{z})q_\varphi(\mathbf{z}|\mathbb{C})}{q_\varphi(\mathbf{z}|\mathbb{C},\mathbb{T})}\right] + \mathbb{E}_{q_\varphi(\mathbf{z}|\mathbb{C},\mathbb{T})}\left[\log \frac{p(\mathbf{z}|\mathbb{C})}{q_\varphi(\mathbf{z}|\mathbb{C})}\right] \tag{15d}$$

$$= -\mathcal{L}_{VI} + \mathbb{E}_{q_\varphi(\mathbf{z}|X_T,Y_T,X_C,Y_C)} \log \frac{p(\mathbf{z}|X_C,Y_C)}{q_\varphi(\mathbf{z}|X_C,Y_C)} \tag{15e}$$

## A.3 GRADIENTS OF THE DIVERGENCE IN NEURAL PROCESSES

We derive the gradients of the KL divergence term in the NP loss. As the the prior model couples its parameters with the posterior model, the result is different from the standard VI.

$$\nabla_\varphi \mathcal{L}_{VI} = \nabla_\varphi \left(\mathbb{E}_{q_\varphi(\mathbf{z}|\mathbb{C},\mathbb{T})}\left[\log \frac{p_\theta(Y_T|X_T,\mathbf{z})q_\varphi(\mathbf{z}|\mathbb{C})}{q_\varphi(\mathbf{z}|\mathbb{C},\mathbb{T})}\right]\right) \tag{16a}$$

$$= \int \left(q_\varphi(\mathbf{z}|\mathbb{C},\mathbb{T})\nabla_\varphi \log \frac{p_\theta(Y_T|X_T,\mathbf{z})q_\varphi(\mathbf{z}|\mathbb{C})}{q_\varphi(\mathbf{z}|\mathbb{C},\mathbb{T})} + \nabla_\varphi q_\varphi(\mathbf{z}|\mathbb{C},\mathbb{T})\left[\log \frac{p_\theta(Y_T|X_T,\mathbf{z})q_\varphi(\mathbf{z}|\mathbb{C})}{q_\varphi(\mathbf{z}|\mathbb{C},\mathbb{T})}\right]\right)d\mathbf{z}$$

$$\tag{16b}$$

$$= \int q_\varphi(\mathbf{z}|\mathbb{C},\mathbb{T})\left[-\nabla_\varphi \log q_\varphi(\mathbf{z}|\mathbb{C},\mathbb{T}) + \nabla_\varphi \log q_\varphi(\mathbf{z}|\mathbb{C})\right] + \nabla_\varphi q_\varphi(\mathbf{z}|\mathbb{C},\mathbb{T})\left[\log \frac{p_\theta(Y_T|X_T,\mathbf{z})q_\varphi(\mathbf{z}|\mathbb{C})}{q_\varphi(\mathbf{z}|\mathbb{C},\mathbb{T})}\right]d\mathbf{z}$$

$$\tag{16c}$$

$$= \int \frac{q_\varphi(\mathbf{z}|\mathbb{C},\mathbb{T})}{q_\varphi(\mathbf{z}|\mathbb{C})}\nabla_\varphi q_\varphi(\mathbf{z}|\mathbb{C}) - \nabla_\varphi q_\varphi(\mathbf{z}|\mathbb{C},\mathbb{T}) + \nabla_\varphi q_\varphi(\mathbf{z}|C,T)\left[\log \frac{p_\theta(Y_T|X_T,\mathbf{z})q_\varphi(\mathbf{z}|C)}{q_\varphi(\mathbf{z}|C,T)}\right]d\mathbf{z} \tag{16d}$$

$$= \int \left(\frac{q_\varphi(\mathbf{z}|\mathbb{C},\mathbb{T})}{q_\varphi(\mathbf{z}|\mathbb{C})}\nabla_\varphi q_\varphi(\mathbf{z}|\mathbb{C}) + \left[\log \frac{p_\theta(Y_T|X_T,\mathbf{z})q_\varphi(\mathbf{z}|C)}{q_\varphi(\mathbf{z}|C,T)} - 1\right]\nabla_\varphi q_\varphi(\mathbf{z}|\mathbb{C},\mathbb{T})\right)d\mathbf{z} \tag{16e}$$

## A.4 DERIVATION OF $\mathcal{L}_{RNP}$ (EQ 7)

$$\min_{\theta,\varphi} D_\alpha\left(q_\varphi(\mathbf{z}|X_T,Y_T,X_C,Y_C)\|p(\mathbf{z}|X_T,Y_T,X_C,Y_C)\right) \tag{17}$$

$$= \min_{\theta,\varphi} \frac{1}{\alpha-1}\log \mathbb{E}_{q_\varphi(\mathbf{z}|X_T,Y_T,X_C,Y_C)}\left[\frac{p(\mathbf{z}|X_T,Y_T,X_C,Y_C)}{q_\varphi(\mathbf{z}|X_T,Y_T,X_C,Y_C)}\right]^{1-\alpha} \quad (\text{By definition}) \tag{18}$$

$$= \max_{\theta,\varphi} \frac{1}{1-\alpha}\log \mathbb{E}_{q_\varphi(\mathbf{z}|X_T,Y_T,X_C,Y_C)}\left[\frac{p(\mathbf{z},Y_T|X_T,X_C,Y_C)}{q_\varphi(\mathbf{z}|X_T,Y_T,X_C,Y_C)}\right]^{1-\alpha} + Const.(\text{Split marginal likelihood})$$

$$\tag{19}$$

$$= \max_{\theta,\varphi} \frac{1}{1-\alpha}\log \mathbb{E}_{q_\varphi(\mathbf{z}|X_T,Y_T,X_C,Y_C)}\left[\frac{p(\mathbf{z},Y_T|X_T,X_C,Y_C)}{q_\varphi(\mathbf{z}|X_T,Y_T,X_C,Y_C)}\right]^{1-\alpha} \quad (\text{Equivalence of removing the constant})$$

$$\tag{20}$$

$$\approx \max_{\theta,\varphi} \frac{1}{1-\alpha}\log \mathbb{E}_{q_\varphi(\mathbf{z}|X_T,Y_T,X_C,Y_C)}\left[\frac{p_\theta(Y_T|X_T,\mathbf{z})q_\varphi(\mathbf{z}|X_C,Y_C)}{q_\varphi(\mathbf{z}|X_T,Y_T,X_C,Y_C)}\right]^{1-\alpha} \tag{21}$$

## A.5 DERIVATION OF $\mathcal{L}_{RNPML}$ (EQ 12)

We start by rewriting the ML objective (Eq 8) as minimizing the KL divergence:

$$- \mathcal{L}_{ML}(\theta) = \max_{\theta} \mathbb{E}_{\mathbb{D}_{\text{train}}} \log p_{\theta}(Y_T|X_T, \mathbb{C}) \tag{22}$$

$$= \max_{\theta} \mathbb{E}_{\mathbb{D}_{\text{train}}} \left[ \frac{1}{N} \sum_{n=1}^{N} \log p_{\theta}(\mathbf{y}_n|\mathbf{x}_n, \mathbb{C}) \right] + Const \quad \text{(Average likelihood for stabilized training)} \tag{23}$$

$$\approx \max_{\theta} \mathbb{E}_{\mathbb{D}_{\text{train}}} \int \hat{p}(\mathbf{y}|\mathbf{x}, \mathbb{C}) \log p_{\theta}(\mathbf{y}|\mathbf{x}, \mathbb{C}) d\mathbf{y} \, (\text{ Definition of the empirical distribution}) \tag{24}$$

$$= \max_{\theta} \mathbb{E}_{\mathbb{D}_{\text{train}}} \int \hat{p}(\mathbf{y}|\mathbf{x}, \mathbb{C}) \left[ \log p_{\theta}(\mathbf{y}|\mathbf{x}, \mathbb{C}) - \log \hat{p}(\mathbf{y}|\mathbf{x}, \mathbb{C}) + \log \hat{p}(\mathbf{y}|\mathbf{x}, \mathbb{C}) \right] d\mathbf{y} \tag{25}$$

$$\equiv \min_{\theta} \mathbb{E}_{\mathbb{D}_{\text{train}}} \left[ D_{\text{KL}} \left( \hat{p}(\mathbf{y}|\mathbf{x}, \mathbb{C}) \| p_{\theta}(\mathbf{y}|\mathbf{x}, \mathbb{C}) \right) \right] \text{(Definition of KLD and removing the constant without } \theta) \tag{26}$$

We now replace the KLD with RD

$$\min_{\theta} \mathbb{E}_{\mathbb{D}_{\text{train}}} \left[ D_{\alpha}(\hat{p}(\mathbf{y}|\mathbf{x}, \mathbb{C}) \| p_{\theta}(\mathbf{y}|\mathbf{x}, \mathbb{C})) \right] \tag{27}$$

$$= \min_{\theta} \mathbb{E}_{\mathbb{D}_{\text{train}}} \frac{1}{\alpha - 1} \left[ \log \int \hat{p}^{\alpha}(\mathbf{y}|\mathbf{x}, \mathbb{C}) p_{\theta}^{1-\alpha}(\mathbf{y}|\mathbf{x}, \mathbb{C}) d\mathbf{y} \right] \text{(Definition of RD)} \tag{28}$$

$$\approx \min_{\theta} \mathbb{E}_{\mathbb{D}_{\text{train}}} \frac{1}{\alpha - 1} \left[ \log \sum_{n=1}^{N} (\frac{1}{N})^{\alpha} p_{\theta}^{1-\alpha}(\mathbf{y}_n|\mathbf{x}_n, \mathbb{C}) \right] \text{(Definition of the empirical distribution)} \tag{29}$$

$$= \min_{\theta} \mathbb{E}_{\mathbb{D}_{\text{train}}} \frac{1}{\alpha - 1} \log \sum_{n=1}^{N} p_{\theta}^{1-\alpha}(\mathbf{y}_n|\mathbf{x}_n, \mathbb{C}) + Const(\text{ Split the non-}\theta \text{ term}) \tag{30}$$

$$\equiv \min_{\theta} \mathbb{E}_{\mathbb{D}_{\text{train}}} \frac{1}{(\alpha - 1)N} \log \sum_{n=1}^{N} p_{\theta}^{1-\alpha}(\mathbf{y}_n|\mathbf{x}_n, \mathbb{C})(\text{Average likelihood for stabilized training}) \tag{31}$$

$$= \min_{\theta} \mathbb{E}_{\mathbb{D}_{\text{train}}} \frac{1}{(\alpha - 1)N} \sum_{n=1}^{N} \log \left( \int p_{\theta}(\mathbf{y}_n, \mathbf{z}|\mathbf{x}_n, \mathbb{C}) d\mathbf{z} \right)^{1-\alpha} \tag{32}$$

## A.6 THEORETICAL RELATIONSHIPS BETWEEN THE $\mathcal{L}_{VI}$, $\mathcal{L}_{ML}$ AND $\mathcal{L}_{RNP}$ OBJECTIVES. 4.2

Our Rényi objective unifies the common three objectives for NPs: $\mathcal{L}_{VI}, \mathcal{L}_{ML}$ (maximum likelihood estimation), and $\mathcal{L}_{CNP}$ (conditional NPs or deterministic NPs).

$$- \mathcal{L}_{RNP} : \frac{1}{1-\alpha} \log \mathbb{E}_{q_{\varphi}(\mathbf{z}|X_T, Y_T, X_C, Y_C)} \left[ \frac{p_{\theta}(Y_T|X_T, \mathbf{z}) q_{\varphi}(\mathbf{z}|X_C, Y_C)}{q_{\varphi}(\mathbf{z}|X_T, Y_T, X_C, Y_C)} \right]^{1-\alpha} \tag{33a}$$

$$- \mathcal{L}_{VI}(\alpha \to 1) : \quad \mathbb{E}_{q_{\varphi}(\mathbf{z}|X_T, Y_T, X_C, Y_C)} \log \left[ \frac{p_{\theta}(Y_T|X_T, \mathbf{z}) q_{\varphi}(\mathbf{z}|X_C, Y_C)}{q_{\varphi}(\mathbf{z}|X_T, Y_T, X_C, Y_C)} \right] \tag{33b}$$

$$- \mathcal{L}_{ML}(\alpha = 0) : \log \mathbb{E}_{q_{\varphi}(\mathbf{z}|X_C, Y_C)} p_{\theta}(Y_T|X_T, \mathbf{z}) \tag{33c}$$

$$- \mathcal{L}_{CNP} (\alpha = 0 \text{ and } q_{\varphi}(\mathbf{z}|X_C, Y_C) = \delta(\varphi(X_C, Y_C)) : \log p_{\theta}(Y_T|X_T, \varphi(X_C, Y_C)) \tag{33d}$$

Proof of the ML objective:

$$\mathcal{L}_{RNP(\alpha=0)} = \log \mathbb{E}_{q_\varphi(\mathbf{z}|X_T,Y_T,X_C,Y_C)} \left[ \frac{p_\theta(Y_T|X_T,\mathbf{z})q_\varphi(\mathbf{z}|X_C,Y_C)}{q_\varphi(\mathbf{z}|X_T,Y_T,X_C,Y_C)} \right] \tag{34a}$$

$$= \log \int q_\varphi(\mathbf{z}|X_T,Y_T,X_C,Y_C) \left[ \frac{p_\theta(Y_T|X_T,\mathbf{z})q_\varphi(\mathbf{z}|X_C,Y_C)}{q_\varphi(\mathbf{z}|X_T,Y_T,X_C,Y_C)} \right] \tag{34b}$$

$$= \log \int p_\theta(Y_T|X_T,\mathbf{z})q_\varphi(\mathbf{z}|X_C,Y_C) = \log \mathbb{E}_{q_\varphi(\mathbf{z}|X_C,Y_C)} p_\theta(Y_T|X_T,\mathbf{z}) = \mathcal{L}_{ML} \tag{34c}$$

Proof of the NPVI objective: Next we will prove that $\mathcal{L}_{RNP,\alpha\to1} = \mathcal{L}_{VI}$. Applying Theorem 5 from (Van Erven & Harremos, 2014a) to the new posteior and prior, we have:

$$\mathcal{D}_{\alpha\to1} \left( q_\varphi(\mathbf{z}|X_T,Y_T,X_C,Y_C) || p(\mathbf{z}|X_T,Y_T,X_C,Y_C) \right) \tag{35a}$$
$$= \mathcal{KL} \left( q_\varphi(\mathbf{z}|X_T,Y_T,X_C,Y_C) || p(\mathbf{z}|X_T,Y_T,X_C,Y_C) \right) \tag{35b}$$

$$= \mathbb{E}_{q_\varphi(\mathbf{z}|X_T,Y_T,X_C,Y_C)} \log \frac{q_\varphi(\mathbf{z}|X_T,Y_T,X_C,Y_C)p(Y_T|X_C,Y_C,X_T)}{p(\mathbf{z},Y_T|X_T,X_C,Y_C)} \tag{35c}$$

$$= -\mathbb{E}_{q_\varphi(\mathbf{z}|X_T,Y_T,X_C,Y_C)} \log \frac{p(\mathbf{z},Y_T|X_T,X_C,Y_C)}{q_\varphi(\mathbf{z}|X_T,Y_T,X_C,Y_C)} + Const \tag{35d}$$

$$\equiv -\mathbb{E}_{q_\varphi(\mathbf{z}|X_T,Y_T,X_C,Y_C)} \left[ \log p(Y_T|\mathbf{z},X_T) + \log p_\varphi(\mathbf{z}|X_C,Y_C) - \log q_\varphi(\mathbf{z}|X_T,Y_T,X_C,Y_C) \right] = \mathcal{L}_{VI} \tag{35e}$$

### A.7 GRADIENTS OF OBJECTIVES FOR RÉNYI NEURAL PROCESSES

$$\nabla \varphi \mathcal{L}_{RNP} = -\nabla \varphi \left( \frac{1}{\alpha-1} \log \int \left[ p(Y_T|X_T,\mathbf{z})^{1-\alpha} q_\varphi(\mathbf{z}|\mathbb{C})^{1-\alpha} q_\varphi(\mathbf{z}|\mathbb{C},\mathbb{T})^\alpha \right] d\mathbf{z} \right) \tag{36a}$$

$$= \frac{1}{1-\alpha} \frac{\int \nabla \varphi \left[ p(Y_T|X_T,\mathbf{z})^{1-\alpha} q_\varphi(\mathbf{z}|\mathbb{C})^{1-\alpha} q_\varphi(\mathbf{z}|\mathbb{C},\mathbb{T})^\alpha \right] d\mathbf{z}}{\int p(Y_T|X_T,\mathbf{z})^{1-\alpha} q_\varphi(\mathbf{z}|\mathbb{C})^{1-\alpha} q_\varphi(\mathbf{z}|\mathbb{C},\mathbb{T})^\alpha d\mathbf{z}} \tag{36b}$$

$$= \frac{1}{1-\alpha} \frac{\int p(Y_T|X_T,\mathbf{z})^{1-\alpha} \nabla \varphi \left[ q_\varphi(\mathbf{z}|\mathbb{C})^{1-\alpha} q_\varphi(\mathbf{z}|\mathbb{C},\mathbb{T})^\alpha \right] d\mathbf{z}}{\int p(Y_T|X_T,\mathbf{z})^{1-\alpha} q_\varphi(\mathbf{z}|\mathbb{C})^{1-\alpha} q_\varphi(\mathbf{z}|\mathbb{C},\mathbb{T})^\alpha d\mathbf{z}} \tag{36c}$$

$$= \frac{1}{1-\alpha} \frac{A}{\int p(Y_T|X_T,\mathbf{z})^{1-\alpha} q_\varphi(\mathbf{z}|\mathbb{C})^{1-\alpha} q_\varphi(\mathbf{z}|\mathbb{C},\mathbb{T})^\alpha d\mathbf{z}} \quad \text{(Product rule)}$$

$$A = \int p(Y_T|X_T,\mathbf{z})^{1-\alpha} \left[ \alpha \left( \frac{q_\varphi(\mathbf{z}|\mathbb{C},\mathbb{T})}{q_\varphi(\mathbf{z}|\mathbb{C})} \right)^{\alpha-1} \nabla_\varphi q_\varphi(\mathbf{z}|\mathbb{C},\mathbb{T}) + (1-\alpha) \left( \frac{q_\varphi(\mathbf{z}|\mathbb{C},\mathbb{T})}{q_\varphi(\mathbf{z}|\mathbb{C})} \right)^\alpha \nabla_\varphi q_\varphi(\mathbf{z}|\mathbb{C}) \right] d\mathbf{z} \tag{36d}$$

### A.8 PRIOR MISSPECIFICATION SETTINGS

We consider two scenarios of prior misspecification: $q(\mathbf{z}|\mathbb{C}_{bad},\varphi)$ and $q(\mathbf{z}|\mathbb{C},\varphi_{bad})$. For $q(\mathbf{z}|\mathbb{C}_{bad},\varphi)$, we design the experiments with the following setting: keep the target data $(X_T,Y_T)$ clean and corrupt the context data with noise $\tilde{y}_\mathbb{C} = (1-\beta) * y_\mathbb{C} + \beta * \epsilon, \epsilon \sim \mathcal{N}(0,1)$. The noise level $\beta$ is set as 0.3 for both training and testing, and the marginal predictive distribution is $p(Y_T|X_T,X_C,\tilde{Y}_C)$ and we report the test target set log-likelihood in Table 4.

For bad parameterization $q(\mathbf{z}|\mathbb{C},\varphi_{bad})$. We adopted domain shift datasets since $p(\mathbf{z}|\mathbb{D}_{\text{train}})$ and $p(\mathbf{z}|\mathbb{D}_{\text{test}})$ do not come from the same distribution. Therefore, the prior model is suboptimal when conditioned on the training parameters $q(\mathbf{z}|\mathbb{C},\varphi_{train})$.

### A.9 AUTOMATIC TUNING OF $\alpha$.

We can start with training the model with the KL objective ($\alpha \to 1$) then gradually decrease (with granularity according to computational constraints) to 0. The intuition is inspired by KL annealing

Table 4: Test log-likelihood with noisy contexts.

| Model | Obj | RBF | Matern 5/2 | Periodic | MNIST | SVHN | CelebA |
|-------|-----|-----|------------|----------|-------|------|--------|
| NP | $\mathcal{L}_{VI}$ | -0.53 ± 0.01 | -0.56 ± 0.01 | -0.74 ± 0.00 | 0.76 ± 0.02 | 2.81 ± 0.04 | 0.89 ± 0.07 |
|    | $\mathcal{L}_{RNP}$ | **-0.45 ± 0.01** | **-0.50 ± 0.01** | **-0.73 ± 0.00** | **0.83 ± 0.02** | **2.98 ± 0.01** | **1.16 ± 0.02** |
| ANP | $\mathcal{L}_{VI}$ | -2.43 ± 0.19 | -2.15 ± 0.18 | **-0.99 ± 0.01** | 0.90 ± 0.02 | 2.83 ± 0.06 | 1.55 ± 0.05 |
|     | $\mathcal{L}_{RNP}$ | **-2.29 ± 0.12** | **-2.11 ± 0.15** | -1.20 ± 0.03 | **0.96 ± 0.01** | **3.11 ± 0.02** | **1.82 ± 0.03** |

Table 5: RNP results with automatic tuning of $\alpha$ values

| Model | Set | Setting | RBF | Matern 5/2 | Periodic | MNIST | SVHN |
|-------|-----|---------|-----|------------|----------|-------|------|
| NP | context | $\mathcal{L}_{VI}$ | 0.69±0.01 | 0.56±0.02 | -0.49±0.01 | 0.99±0.01 | 3.24±0.02 |
|    |         | $\mathcal{L}_{RNP\_ada\alpha}$ | **0.75±0.02** | **0.61±0.02** | **-0.49±0.00** | **1.01±0.01** | **3.26±0.01** |
|    | target | $\mathcal{L}_{VI}$ | 0.26±0.01 | 0.09±0.02 | **-0.61±0.00** | 0.90±0.01 | 3.08±0.01 |
|    |        | $\mathcal{L}_{RNP\_ada\alpha}$ | **0.31±0.01** | **0.13±0.01** | **-0.61±0.00** | **0.92±0.01** | **3.10±0.01** |
| ANP | context | $\mathcal{L}_{VI}$ | **1.38±0.00** | **1.38±0.00** | 0.65±0.04 | **1.38±0.00** | **4.14±0.00** |
|     |         | $\mathcal{L}_{RNP\_ada\alpha}$ | **1.38±0.00** | **1.38±0.00** | **0.97±0.11** | **1.38±0.00** | **4.14±0.00** |
|     | target | $\mathcal{L}_{VI}$ | 0.81±0.00 | 0.64±0.00 | -0.91±0.02 | **1.06±0.01** | **3.65±0.01** |
|     |        | $\mathcal{L}_{RNP\_ada\alpha}$ | **0.83±0.01** | **0.66±0.01** | **-0.71±0.05** | **1.06±0.01** | **3.65±0.01** |
| TNPD | context | $\mathcal{L}_{VI}$ | **2.58±0.01** | **2.57±0.01** | **-0.52±0.00** | 1.73±0.11 | 10.63±0.12 |
|      |         | $\mathcal{L}_{RNP\_ada\alpha}$ | **2.58±0.01** | **2.57±0.01** | **-0.52±0.00** | **1.94±0.02** | **10.73±0.57** |
|      | target | $\mathcal{L}_{VI}$ | 1.38±0.01 | **1.03±0.00** | **-0.59±0.00** | **1.63±0.07** | 6.69±0.04 |
|      |        | $\mathcal{L}_{RNP\_ada\alpha}$ | **1.39±0.00** | **1.03±0.00** | **-0.59±0.00** | 1.56±0.02 | **6.71±0.24** |

Table 6: Simple baseline comparison. In the setting column Separate PQ means the prior and posterior models are parameterised separately for the NP frameworks and optimised using the VI objective

| Model | Set | Setting | RBF | Matern 5/2 | Periodic | MNIST | SVHN |
|-------|-----|---------|-----|------------|----------|-------|------|
| NP | context | Separate PQ | 0.56±0.02 | 0.41±0.01 | -0.50±0.00 | 1.00±0.03 | 3.21±0.01 |
|    |         | $\mathcal{L}_{RNP(\alpha)}$ | **0.78±0.01** | **0.66±0.01** | **-0.49±0.00** | **1.01±0.02** | **3.26±0.01** |
|    | target | Separate PQ | 0.18±0.01 | 0.01±0.00 | **-0.61±0.00** | 0.90±0.02 | 3.05±0.01 |
|    |        | $\mathcal{L}_{RNP(\alpha)}$ | **0.33±0.01** | **0.16±0.01** | -0.62±0.00 | **0.91±0.01** | **3.09±0.01** |
| ANP | context | Separate PQ | **1.38±0.00** | **1.38±0.00** | -0.17±0.25 | **1.38±0.00** | **4.14±0.00** |
|     |         | $\mathcal{L}_{RNP(\alpha)}$ | **1.38±0.00** | **1.38±0.00** | **1.22±0.02** | **1.38±0.00** | **4.14±0.00** |
|     | target | Separate PQ | 0.80±0.01 | 0.63±0.01 | -0.70±0.02 | **1.06±0.00** | **3.66±0.01** |
|     |        | $\mathcal{L}_{RNP(\alpha)}$ | **0.84±0.00** | **0.67±0.00** | **-0.57±0.01** | 1.05±0.01 | 3.61±0.02 |

for VAE models Bowman et al. (2016), which starts with a strong prior penalization (close to 1) to reduce the posterior variance quickly and gradually reduces the prior penalization (close to 0) and focuses more on model expressiveness. The results are presented in Table 5. Our heuristics still managed to outperform the baselines across multiple datasets and methods.

## A.10 ADDITIONAL RESULTS.

We provided some qualitative results of the baseline NPs as well as their corresponding RNPs.

Table 6 compares RNP with a simple baseline model using separate parameters of the prior and posterior models.

Table 7 added a wall clock time comparison between our objective and the VI objective.

**Alternative objective using a prior-based "local" RD.** Recall that in standard variational inference the main goal is to minimize the KL divergence between the approximate posterior and the true posterior $\mathcal{KL}(q(\mathbf{z}|\mathbf{x})||p(\mathbf{z}|\mathbf{x}))$. Since the true posterior is unavailable, VI instead maximizes the ELBO, which is the sum of an expected log likelihood term and the negative KL divergence between the approximate posterior and the prior $\mathcal{KL}(q(\mathbf{z}|\mathbf{x})||p(\mathbf{z}))$.

In contrast, our approach minimizes the RD directly, as it involves the logarithm of an expectation instead of the expectation of a logarithm. However, it is possible to maximize an objective analogous

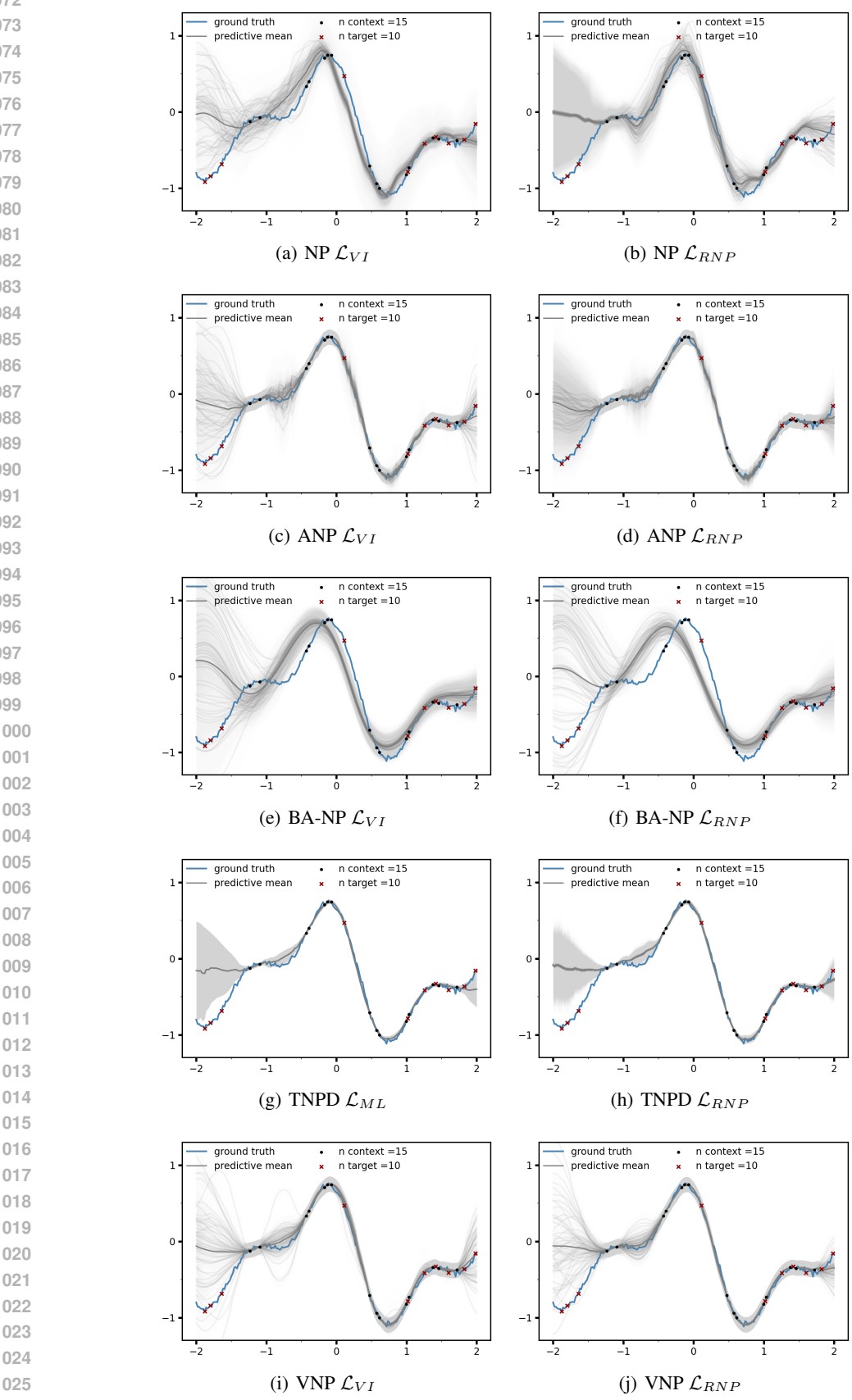

Figure 6: 1D regression RBF dataset.

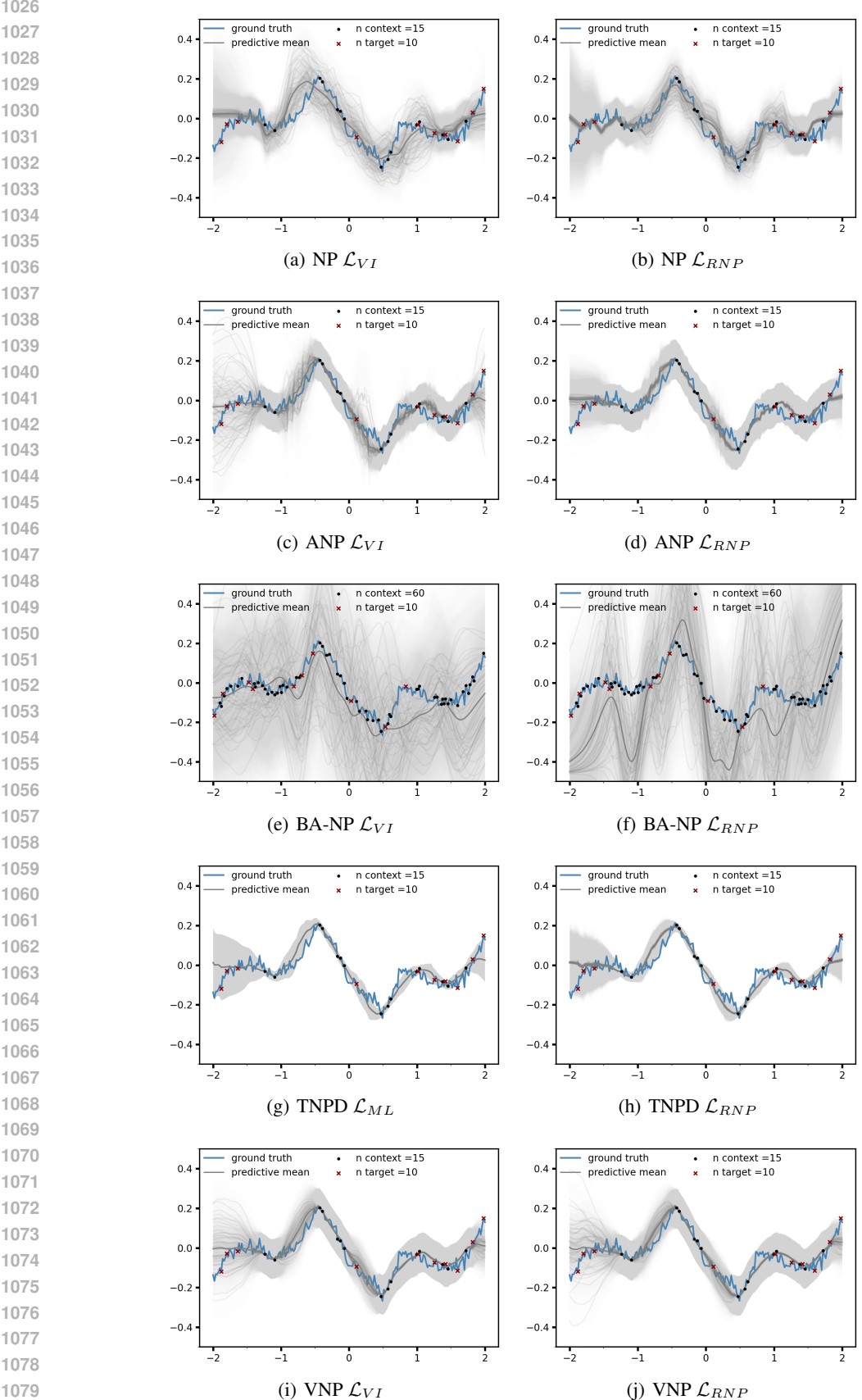

Figure 7: 1D regression Matern dataset.

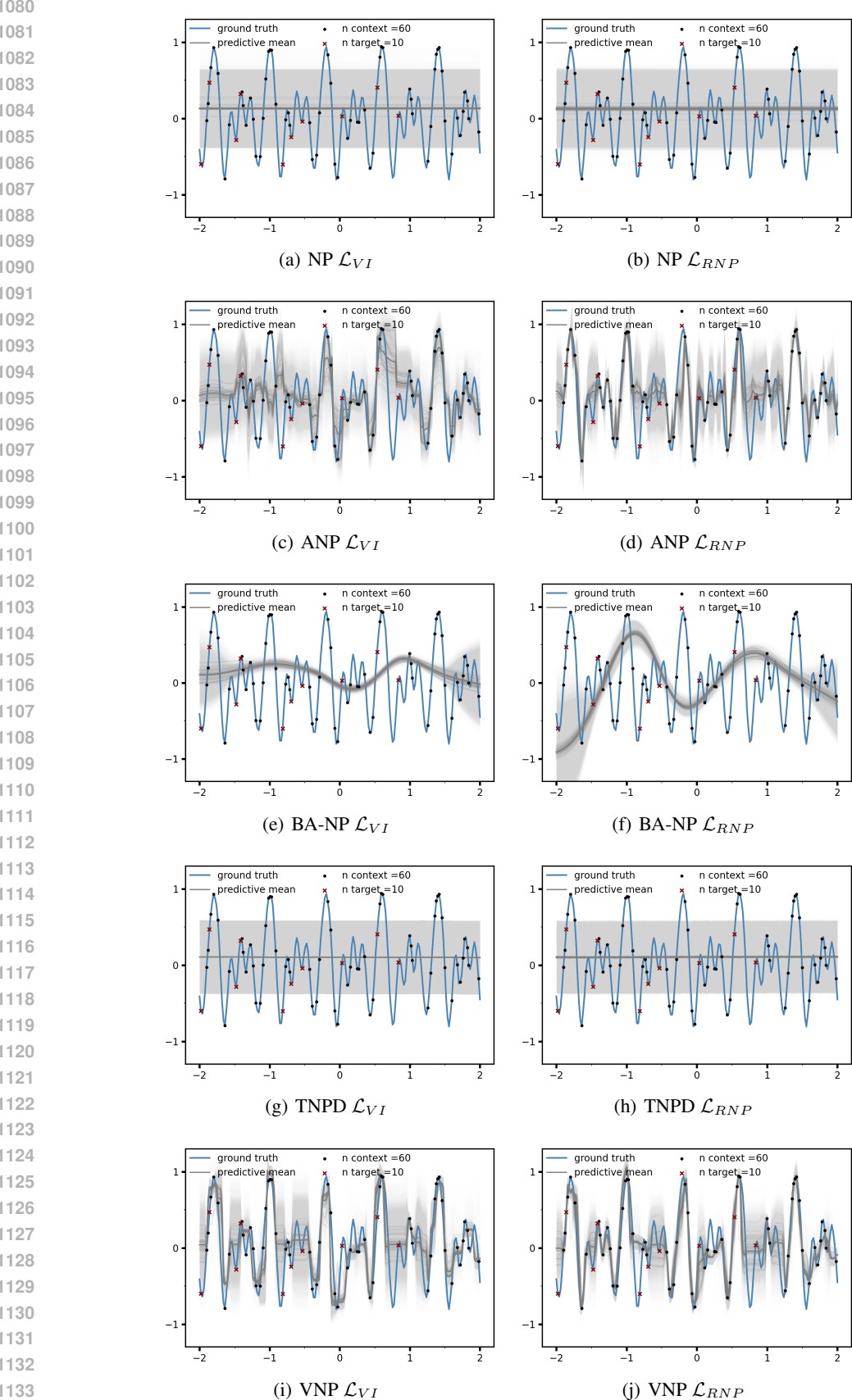

Figure 8: 1D regression Periodic dataset.

Table 7: Wall clock time (in seconds) comparison. 32 MC samples were chosen during training to be consistent with the results in Table 1.

| Model | Setting | RBF | Matern 5/2 | Periodic | MNIST | SVHN |
|---|---|---|---|---|---|---|
| NP | $\mathcal{L}_{VI}$ | 1028 | 1015 | 1067 | 3231 | 4620 |
| | $\mathcal{L}_{RNP(\alpha)}$ | 1108 | 1000 | 1048 | 3058 | 4780 |
| ANP | $\mathcal{L}_{VI}$ | 1585 | 1620 | 1721 | 3770 | 5391 |
| | $\mathcal{L}_{RNP(\alpha)}$ | 1712 | 1693 | 1671 | 3605 | 5444 |

to the ELBO: $\mathbb{E}_{q(\mathbf{z}|X_T,Y_T,X_C,Y_C)} \log p(Y_T|X_T,\mathbf{z}) - \mathcal{D}_\alpha(q(\mathbf{z}|X_T,Y_T,X_C,Y_C)||p(\mathbf{z}|X_C,Y_C))$ that replaces the prior-based KL divergence with the RD. We will refer to this objective as "local".

Fig 9 shows the test log-likelihood difference between two objectives using the optimal $\alpha$ values. Our method outperforms the local Rényi divergence on three GP regression tasks. By comparing the gradient over the posterior parameters of the two objectives, our objective adds a scaling factor $p(Y_T|X_T,\mathbf{z})^{1-\alpha}$ to the "local" RD objective, which imposes a larger gradient penalty on less confident predictions and therefore focus more on samples with low likelihoods. This further improves the mass covering behaviour of the RNP and leads to better log-likelihood values.

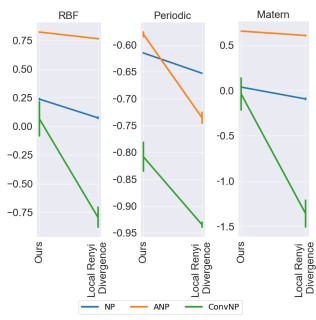

Figure 9: Local $\alpha$ divergence