# OpenReview forum: "Rényi Neural Processes"
_ICLR.cc/2025/Conference — Submitted to ICLR 2025_

### Official Review · Reviewer_6B6t · 2024-10-29

**Soundness:** 2
**Presentation:** 3
**Contribution:** 2
**Rating:** 6
**Confidence:** 4

**Summary:**

The paper introduces Rényi Neural Processes (RNP), a simple method that replaces KL divergence (KLD) with Rényi divergence (RD) to improve the neural process (NP) when there is prior misspecification. They propose RD-based objective functions for both commonly used NP objectives - Variational Inference (VI) in latent NP and Maximum Likelihood (ML) in conditional NP. The authors also provide theoretical analysis and proofs to support their proposed objective functions. In the experimental section, they implement RNP across multiple NP variants and demonstrate consistent improvements over the original models.

**Strengths:**

* The proposed method is simple yet effective, and can be easily deployed to other NP models by simply replacing the corresponding objective functions. This makes the method highly practical.
* The paper is overall well-written, with thorough literature review and clear, intuitive figures that effectively convey the key concepts.
* The paper employs a comprehensive set of baselines covering both CNP and NP variants, including the current state-of-the-art TNP method. The evaluation is also conducted on two real-world datasets, showing its practical applicability. Besides, the authors provide guidance for choosing $\alpha$, along with several ablation studies.
* They provide theoretical analysis connecting VI and ML objectives through the $\alpha$ parameter, and prove how RNP helps with prior misspecification.
* Their approach to rewriting the ML estimation as minimizing the KLD between the empirical distribution and the model distribution, and then applying RD is an interesting perspective.

**Weaknesses:**

I am still trying to understand the definition of prior misspecification in your paper, it seems to lack clarity and consistency, and there appears to be a disconnect between the theoretical motivation and experimental validation.

Based on the introduction and the definition of RD, prior misspecification is presented as the mismatch between context and target sets, which commonly occurs when collecting context data with additional noise or other uncontrollable factors.
However, the toy examples and experimental setups seem to address a different type of prior misspecification - the mismatch between training and test distributions ($D_{train}$ and $D_{test}$), which is more akin to domain shift problems.

These are arguably two distinct scenarios, and the paper doesn't clearly explain:

* How these two types of misspecification relate to each other
* Why RD would help with the domain shift scenario
* Whether these can be unified under a single framework of prior misspecification

If I have misunderstood any points, I would appreciate clarification from the authors, and I am willing to increase my score if my concerns are fully addressed.

Some other points:
* In Section 7.1, the experiments are conducted on seemingly well-specified datasets, yet the RD-based methods still outperform the baselines. Is there any justification explains these improvements?
* Line 229, ANP uses attention that learns to attend to the contexts relevant to the given target, to my understanding they didn't incorporate dependencies between target points, maybe you are talking about GNP[1] which models the target points jointly?
* Line 242, TNP has three versions, you are talking about TNP-A (autoregressive version), for example, TNP-D doesn't do any autoregressive prediction. And I think you are using TNP-D in the experiment section.
* Line 311, isn't the posterior distribution be $p(z|X_t, Y_t, X_c, Y_c)$ as you mentioned in eq.3?
* Line 352, TNP has an inappropriate reference, Maraval et al., 2024 only apply TNP to the BO setting.

[1] Markou, S., Requeima, J., Bruinsma, W., Vaughan, A., & Turner, R. E. Practical Conditional Neural Process Via Tractable Dependent Predictions. In International Conference on Learning Representations.

**Questions:**

* In Table 1, are you using ML or VI objectives for your RD-based loss functions?
* I am not very sure how and why you are calculating the log-likelihood on the context set. What data are you conditioning on when predicting the context set?
* On TNP-D, the results using $L_{RNP}$ are always better than using $L_{ML}$ in Table 1, why on the EMNIST dataset (Table 2), the results become worse?

---

> ### Author Response · Authors · 2024-11-22
>
> We thank the reviewer for acknowledging our theoretical novelties in the ML-based objectives and the comprehensive experiments.
>
> # Prior misspecification motivation and experiment validation.
>
> The prior misspecification defined in Proposition 3.2 is the mismatch between the  model prior distribution $q(\mathbf{z}|C, \varphi)$ which is conditioned on BOTH the parameters $\varphi$ and context data ${C}$ and the ground truth prior distribution $p(\mathbf{z}|\{C}, \varphi^*)$ whose ground truth parameters $\varphi^*$ are unknown (which are not the training/testing distributions the reviewer was mentioning). We can risk obtaining a misspecified prior by conditioning either on a poor parameterization or poor data, which is what we've covered in the experiment Section 7.1 and section 7.2.
>
> Scenario 1: Poor parameterization $q(\mathbf{z}|{C}, \varphi_{bad})$   (Section 7.1)
>
>  meaning NN parameters $\varphi_{bad}$ are far from the ground truth values $\varphi^*$. This can happen in NPs because their posterior and prior models are forced to couple the parameters, which is why we saw performance improvements on Table 1 with the Renyi divergence even when there are no domain shifts.
>
> Scenario 2: Poor data $q( \mathbf{z}|{C}_{bad}, \varphi*)$ (Section 7.2)
>
>  meaning the data $C_{bad}$ are sampled from a different distribution than training. This, as the reviewer pointed out, can be caused by additional noise or uncontrollable factors. We used the domain shift data sets Lotka-Volterra and EMNIST to validate the performance of the model, as we can ensure that the data distribution is different, and the results in Table 2 have validated the superiority of the RNPs in that regard. Our framework can indeed handle both types of misspecification with the Renyi Divergence by mitigating the impact of the prior model $q(z|C_{bad}, \varphi_{bad})$ on the posterior and likelihood model with the alpha value and it does not distinguish the two scenarios. We have added clarifications in both the definition and experiments.
>
> # Other points.
>
> 1.  Target dependencies in ANP. Target sets are indeed assumed independent in ANPs and their attention is computed within the context set or between the context and each target point. We have moved this part to the posterior model for introducing dependencies. We also added the introduction of dependencies of GNPs in the ML objective section.
>
> 2. Autoregressive TNP. We have clarified TNP-A to be the model which incorporate dependencies.
>
> 3. Posterior typo. We have corrected the typo to  $q(\mathbf{z}|X_T, Y_T, X_C, Y_C)$.
>
> 4. Reference. We have referred to the original TNP paper.
>
> # Q1 VI/ML objective for RNPs.
>
> We adopted the VI objectives for NPs, ANPs and VNPs (Versatile Neural Processes) as their model designs include the prior models. We used the ML objective for TNP and BANP(Bayesian aggregation NPs) because the TNP objective was originally defined using ML only and the ML objective significantly outperformed the VI objective for BANPs. We have clarified the setting in the experiment section.
>
> # Q2 Reporting context log-likelihood.
>
> Our reported value is based on the overall predictive distribution including both the context and target sets:  $p(Y_T, Y_C|X_T, C)$.
> It's important to distinguish between context and target log-likelihoods because, during training, Neural Process models can sometimes improve overall log-likelihood by sacrificing one of these aspects. Similar to [3,4], separately checking context and target log-likelihoods would provide more insights.
>
> [3] J. Lee, Y. Lee, J. Kim, E. Yang, S. J. Hwang, and Y. W. Teh. Bootstrapping neural processes. In Advances in Neural Information Processing Systems 33 (NeurIPS 2020), 2020.
>
> [4] H. Lee, E. Yun, G. Nam, E. Fong, and J. Lee. Martingale posterior neural processes. In International Conference on Learning Representations (ICLR), 2023.
>
> # Q3 EMNIST dataset performance.
>
> No hyperparameter tuning was adopted on the alpha value, which is possible to obtain suboptimal results than the original ML objective. EMNIST dataset seems to produce a higher variance in the test log-likelihood 1.47±0.12 than the datasets in Table I (usually ~0.0x) which could contributes to the worse results in the testing set with no domain shifts.

---

> > ### Comment · Reviewer_6B6t · 2024-11-25
> >
> > Thank you for your response which has addressed several of my concerns.
> >
> > I still have some questions about the explanation of prior misspecification (Answer 1):
> >
> > Regarding scenario 1, my current understanding of prior misspecification (based on definition 3.1) is that $q_\varphi(z|C)$ cannot match the true prior $p(z|C)$ under any parameterization. However, I still couldn't understand why would parameter coupling between prior and posterior models necessarily result in the inability to match the true prior distribution, even when the data distribution is well-specified in section 7.1?
> >
> > For the scenario 2, I still have some confusion between two distinct types of data misspecification:
> > * Context-Target mismatch: where observed context data differs from prediction targets (e.g., context data is noisy, target data is clean)
> > * Train-Test distribution shift: where training and test data come from different distributions, but there is no mismatch between contexts and targets.
> >
> > The Lotka-Volterra experiment validates RNP's effectiveness under Train-Test distribution shift, but this is conceptually different from Context-Target mismatch, as context and target data are from the same distribution. For Train-Test distribution shift, the problem might be $\varphi$ is optimal to training set, but not optimal to test set, which I think should be categorized into $\varphi_{bad}$ case. I suggest that authors explain this distinction more clearly, given that prior misspecification has never been properly defined in NP literature.
> >
> > For other questions and points, thanks for clarification!

---

> ### Author Response · Authors · 2024-11-26
>
> We thank the reviewer for clarifying the questions for discussion.
>
> # Scenario 1
> The objective of NP in eq(3) suggests that by coupling the parameters $\varphi$ the model can minimize $KL(q(z|C, T, \varphi) \mid q(z|C, \varphi))$, for example, by moving the prior $q(z|C, \varphi)$ closer to the posterior $q(z|C, T, \varphi)$, but it doesn't guarantee the minimization of $KL(q(z|C, T, \varphi)\mid p(z| C, \varphi^*))$ which corresponds to maximizing the true ELBO.
>
> # Scenario 2
> We have added a context-target mismatch experiment in Supplementary Table 4. The setting is detailed as follows: during both training and testing, we keep the target data $(X_T, Y_T)$ clean and corrupt the context data with noise $\tilde{y} = (1-\beta) * y_C + \beta * \epsilon,  \epsilon \sim \mathcal{N}(0, 1)$. The noise level $\beta$ is set as 0.3 for both training and testing. Therefore, the marginal predictive distribution is $p(Y_T|X_T, X_C, \tilde{Y})$. This can indeed be categorized as the poor context $p(z|C_{bad}, \varphi)$. Our method still showed outperformance across multiple datasets as the impact of misspecified contexts is alleviated via the divergence.  Along with the train-test distribution shift scenario, which the reviewer believes to be categorized as bad parameterization, we have shown improvements in both scenarios. We have added relevant clarifications in section 7.2 to distinct the two scenarios.
>
> | Model  |      Obj           |           RBF                | Matern 5/2       | Periodic         | MNIST           | SVHN            | CelebA          |
> |:-------------:|:-------------:|:-------------:|:----------------:|:----------------:|:----------------:|:----------------:|:----------------:|
> | NP    | $\mathcal{L}_{VI}$  | -0.53   $\pm$ 0.01 | -0.56 $\pm$ 0.01 | -0.74 $\pm$ 0.00 | 0.76 $\pm$ 0.02 | 2.81 $\pm$ 0.04 | 0.89 $\pm$ 0.07 |
> |       | $\mathcal{L}_{RNP}$ | **-0.45 $\pm$ 0.01**   | **-0.50 $\pm$ 0.01** | **-0.73 $\pm$ 0.00** | **0.83 $\pm$ 0.02** | **2.98 $\pm$ 0.01** | **1.16 $\pm$ 0.02** |
> | ANP   | $\mathcal{L}_{VI}$  | -2.43 $\pm$ 0.19   | -2.15 $\pm$ 0.18 | **-0.99 $\pm$ 0.01** | 0.90 $\pm$ 0.02 | 2.83 $\pm$ 0.06 | 1.55 $\pm$ 0.05 |
> |       | $\mathcal{L}_{RNP}$ | **-2.29 $\pm$ 0.12**   | **-2.11 $\pm$ 0.15** | -1.20 $\pm$ 0.03 | **0.96 $\pm$ 0.01** | **3.11 $\pm$ 0.02** | **1.82 $\pm$ 0.03** |
>
> Hopefully our responses have helped address your concerns. We are more than happy to provide further discussions if needed.

---

> > ### Author Response · Authors · 2024-11-27
> >
> > We greatly appreciate the reviewer’s valuable feedback, which has been very helpful throughout this process. We would like to confirm whether our responses have satisfactorily addressed their additional questions. Given this and the reviewer’s acknowledgment that our earlier responses addressed their initial concerns, we kindly ask whether the reviewer might consider revising their original score.

---

> > > ### Comment · Reviewer_6B6t · 2024-11-27
> > >
> > > Thank you for your additional explanation and experiments. Based on our discussion, my understanding of your definition of prior misspecification is that $p(z|C) \neq q(z|C,\varphi)$. I still find this somewhat counterintuitive as it means that all NP methods inherently face this issue even when the data is from the same distribution. Nevertheless, given your thorough responses, additional experiments, and clarification of most of my concerns, I am happy to increase the score to 6.

---

### Official Review · Reviewer_JR2F · 2024-10-31

**Soundness:** 4
**Presentation:** 3
**Contribution:** 3
**Rating:** 8
**Confidence:** 4

**Summary:**

This paper aims to address a key potential weakness in latent variable neural processes (NP), namely, that the true prior $p(\mathbf{z} \mid X_C, Y_C)$ can be approximated by a parametric model $q_\phi(\mathbf{z} \mid X_C, Y_C)$ which shares parameters with the approximate posterior model. When the prior is “misspecified”, as defined rigorously in the text, the variational bound used to optimize NP is no longer valid and the authors argue leads to worse data fitting. They remedy this issue by replacing the KL divergence in the lower bound with a Rényi divergence, whose hyperparameter $\alpha$ enables more expressive posteriors. On a variety of experiments commonly used in the NP literature, the authors demonstrate that existing NP models benefit from the Rényi divergence variation of the objective.

**Strengths:**

The authors identify an important modeling assumption in NP that can be a source of failure for these models and suggest an easy to implement and intuitive remedy.

The experimental section is well laid out, and I appreciated that the authors explicitly connected each experimental subsection to an empirical question, e.g., in lines 397-400. The empirical results themselves are extensive. The improvements over strong baselines, such as TNP, are compelling.

**Weaknesses:**

The list below roughly follows the order of appearance in the manuscript. In my opinion W2, W6, W10 are the most important among these.

### W1: The claims at the end of the introduction seem to overlap quite a bit.
For example, in contribution 1, the authors already discuss the improved likelihoods, which seems like it is more a part of the empirical contributions of 2 and 3. Additionally, again, in contribution 2 the reference to “consistent log-likelihood improvements” is actually the main point of the contribution of the third bullet.

### W2: The effects of the misspecified prior can be made more rigorous
Currently, the authors claim that KL divergence leads to:
- Biased estimation of the posterior variance
- Less expressive models that underfit

However, analyzing these claims analytically would be quite helpful ,e.g., what is the analytical form of this bias in estimating the posterior variance and is there a way to make the “underfitting” more rigorous. Currently, the main analysis is a single motivating example. For example, Figure2b is quite anecdotal and only shows a single sample.

I recognize that this is not a small ask, but I believe it will significantly improve the quality of Section 2 and the overall motivation.

### W3: Notation of Definition 3.1 seems detached from the rest of the paper.
How does $\eta$ relate to the rest of the notation introduced in this work? Does it correspond to a shared parametric form of $p_\varphi$ and $q_\varphi$? If so, this needs to be spelled out.

### W4: Proposition 3.2 is missing an “if”
Shouldn’t there be an “if” in lines 173 and 174:
> …, **if** the prior model is misspecified,...

Currently the proposition reads as if misspecification is inevitable, which, in my understanding, is not necessarily true.

### W5: I think the equation for $\mathcal{RNP}$ is imprecise
You are taking a Monte Carlo estimate of an expectation that relies on the parameters over which you are optimizing (presumably via stochastic gradient descent). This would require some methodology along the lines of the reparameterization trick which should be explicitly stated here.

### W6: Prior misspecification does not seem related to the ML-based objective
It is not clear to me from the current exposition why prior misspecification should affect the ML-based version of the NP objective.

### W7: Lines 242-247 seem out of place / irrelevant
I do not understand how these lines relate to the rest of this section. The generalization / recasting of TNP-A as an implicit latent variable model with Diracs is not used in the rest of the exposition / derivation to the best of my understanding.

### W8: The tabular regression experiments (Pace & Barry, 1997) appear to be missing
These results are not in the experimental results sections nor in any of the appendix sections

### W9: The details for the differential equations (e.g. Hare-Lynx and Lotka-Volterra) should be added to Section 7 before the Baselines paragraph.
A similar level of detail to that provided for GP/inpainting experiments should be added for these DE experimental setups as well.

### W10: The improvements seen in Table 1 (i.e., where misspecification is presumably not a problem) are not well explained.
Why do we see gains in the standard GP / inpainting experiments from RNP if there is not necessarily a misspecification issue here?

### W11: The additional compute overhead from the Monte Carlo approximation should be quantified.
Both for the main results and in the ablation analysis, the authors should have a secondary axis or some way of conveying how the actual compute overhead and how it grows in $K$.

### W12: The graphs are quite small and difficult to read without extensively zooming.
The legends are quite difficult to make out.

Additionally, Figure3b has a different line color for some reason and Figures 4a and 4b are missing a legend for the red dot indicator.

### W13: Typos / Grammatical Errors
Below I list the minor typos/errors that I noticed through a preliminary read:
- Line 035: “advance” should perhaps be “advantage”?
- Line 061: “poster variance” should be “posterior”
- Line 074: “achieve the better…” should be “achieve a better”
- Lines 149-150: The wording is confusing here “the model can reduce the prior penalty less than significantly than the standard KL”
- Line 152: “overestimate” should be “overestimated”
- Line 216: There is a missing space between “inA.3”
- Line 233: $p_\varphi$ should be $q_\varphi$, I believe.
- Line 239: $p_\varphi$ should be $q_\varphi$, I believe.
- Line 269: $d\mathbf{z}$ should be removed from equation 12.
- Lines 404-405: These lines are written in a slightly confusing way. It sounds like the $\alpha$ value is what the baseline corresponds to. But the baselines use $\alpha = 0$ / $\alpha=1$ right?
- Lin 783: Should be $\alpha \rightarrow 1$ in the parentheses
- Table 1,Table 2, and Table 3 captions have duplicated $\uparrow$’s and $\downarrow$’s

### W14: Move Table 4 to the start of the appendix.
Not a real weakness, just a suggestion. Feel free to ignore it.

**Questions:**

### Q1: Gradient of Renyi divergence is potentially misleading
Is the argument here potentially incorrect/misleading since the the gradient needs to also be taken with respect to the $q_\varphi(\mathbf{z\mid X_C, Y_C) \approx p(\mathbf{z})$?

### Q2: In A.3, doesn’t the logic from Eq 20 to 21 require that the prior is well specified?
Does this equality hold without this assumption?

---

> ### Author Response · Authors · 2024-11-22
>
> We deeply appreciate the reviewer's acknowledgment with our experimental validation and outperformance. We will start by addressing the more important questions.
>
> # W2 Analytical analysis of prior misspecification.
>
> We can write the gradient of the objective (6) wrt the parameters of the posterior as
>
> $ \mathcal{L} =  \frac{1}{1-\alpha} \log E_{q(\mathbf{z}|C, T, \varphi )} \left[\frac{p(Y_T| X_T, \mathbf{z}, \theta) q(\mathbf{z}|C, \varphi)}{q(\mathbf{z}|C, T, \varphi)}\right]^{1-\alpha}$
>
>
> $ \nabla_\varphi \mathcal{L} =  E_{q(\mathbf{z}|C, T, \varphi)}  \left[ w_\alpha(\mathbf{z}, X_T, Y_T, C, ) \nabla_\varphi \log \frac{p(Y_T|X_T, \mathbf{z}, \theta) q(\mathbf{z}| C, \varphi)}{q(\mathbf{z}|C, T, \varphi)}\right]  $
>
>    with $ w_\alpha(\mathbf{z}, X_T, Y_T, C)  =  \left(\frac{p(Y_T|X_T, \mathbf{z}, \theta) q(\mathbf{z}|C, \varphi)}{q(\mathbf{z}|C, T, \varphi)}  \right) ^{1-\alpha}  / {E}_{q(\mathbf{z}|C, T)}  \left[ \frac{p(Y_T|X_T, \mathbf{z}, \theta) q(\mathbf{z}|C, \varphi)}{q(\mathbf{z}|C, T, \varphi)}\right]^{1-\alpha}  $
>
>
> $w_\alpha(\mathbf{z}, X_T, Y_T, C)$ is the normalized importance weights. Therefore, the gradient of RNP equals the gradients of the VI objective scaled by the weight  $w_\alpha(\mathbf{z}, X_T, Y_T, C)$. If the prior model $p_\varphi(\mathbf{z}|C)$ is misspecified, by selecting different $\alpha$ values, we can control the behavior the posterior update which consequently affect the posterior variance and mitigate prior misspecification. In a simple case of a Bayesian linear regression model $y= \mathbf{\theta}^T\mathbf{x} + \epsilon$ with two dimensional inputs and one dimensional output, the analytical solutions of the posterior is a mutilvariate Gaussian $p(\theta|\mathbf{x}, y) \sim \mathcal{N}(\mu, \Sigma)$. Suppose we use a factorized Gaussian as an approximate posterior $q(\theta)=\prod_i
>  q(\theta_i)$ with $q(\theta_1) = \mathcal{N}(\mu_1, \lambda_1^{-1})$, we can show the variance of the approximate posterior factorized Gaussian varies by alpha:
>  $\lambda_1 = \rho_\alpha \Sigma_{11}$
>  where $\rho_\alpha = \frac{1}{2\alpha}\left[ (2\alpha -1) + \sqrt{ 1- \frac{4 \alpha(1-\alpha)\Sigma^2_{12}}{\Sigma_{11} \Sigma_{22}}}\right]$ is non-decreasing in $\alpha$.
>
>
>
>
> # W6 Prior misspecification in ML objectives
> The prior can still be misspecified in NN parameters of $p_\varphi(\mathbf{z}|C)$ and the family of distributions we choose for the approximation $p$. For more detailed explanation of limitations on the approximation, please refer to Section 3 likelihood approximation in [1].
>
> [1] Volpp, Michael, et al. "Bayesian Context Aggregation for Neural Processes." ICLR. 2021.
>
> # W10 Performance improvements in Table 1.
>
> Please refer to the comment we left for Reviewer 6B6t (Prior misspecification motivation and experiment validation, senario 1) regarding the bad parameterization in the neural networks.
>
>
> # Q1 Misleading gradients.
> The second term in Eq (13 (b)) did consider the gradient wrt to $q_\varphi(\mathbf{z}|\mathbb{C})$ which adjust the prior to be closer to the posterior as well.
>
> # Q2 A.3 Well specification.
> The logic is still to couple the parameters between the posterior and prior model which approximates the ground truth prior. However, RNP dampens the effect of the bad approximation so that the model focuses more on improving the likelihood.
>
>
> # Other issues.
>
> W1. We have shortened the relevant contribution in the introduction section.
>
> W3. We have added the clarification: ``This translate to NPs as the approximate prior model $q_\varphi(\mathbf{z}|X_C, Y_C)$ in Eq 3 can not recover the ground truth prior $p(\mathbf{z}|X_C, Y_C)$ for any parameterization of $\varphi$ ''.
>
> W4. We have revised the proposition 3.2 accordingly.
>
> W5. We have added the reparameterization trick for computing the SGD over the parameters of the posterior.
>
> W7. We have removed the generalization of TNP-A with Diracs in the section.
>
> W8. We have removed the redundant ``tabular regression'' term.
>
> W9. We have added the ODEs for the Lotka-Volterra datset and elaborated on the parameter setting of the dynamics and NP setting for training.
>
> W11. Please refer to the comment we left for Reviewer eR5B regarding MC efficiency.
>
> W12. We have increased the legend size and the figure size. We kept the consistent color and added the legend for red dot indicator.
>
> W13. We have fixed the typos and confusing wordings.
>
> W14. We have moved the notation table to the start of the appendix.

---

> > ### Comment · Reviewer_JR2F · 2024-11-25
> > **Response to rebuttal**
> >
> > Thank you for addressing my concerns and the response to my question. I maintain my score of 8 which reflects that this is good work (that shows clear improvement over baselines) worthy of acceptance.

---

### Official Review · Reviewer_eR5B · 2024-11-02

**Soundness:** 3
**Presentation:** 4
**Contribution:** 3
**Rating:** 3
**Confidence:** 3

**Summary:**

The paper introduces Renyi Neural Process, a framework that addresses the limitations of standard Neural Processes (NPs) by mitigating prior misspecification through a new objective function. RNP enhances uncertainty estimation and robustness in learning by applying Rényi divergence instead of traditional KL divergence, leading to improved performance across various benchmarks. The authors demonstrate consistent log-likelihood improvements in tasks such as 1D regression and image inpainting while acknowledging the trade-off between computational efficiency and performance due to Monte Carlo sampling.

**Strengths:**

RNP consistently achieves better log-likelihood and mitigates over-smoothing in predictions, particularly in challenging tasks like periodic data regression and higher-dimensional image inpainting. By utilizing Renyi divergence, RNP enhances the expressiveness of the posterior distribution, leading to more reliable uncertainty quantification compared to traditional Neural Processes.

**Weaknesses:**

The use of Renyi divergence requires additional computations, particularly due to the need for Monte Carlo sampling to estimate the divergence. This can lead to longer training times and higher resource consumption compared to standard Neural Processes, which may limit scalability in large datasets or real-time applications. RNP introduces extra parameters that control the behavior of the divergence, which can make the model sensitive to hyperparameter tuning. Improper selection of these parameters may lead to suboptimal performance, requiring extensive experimentation to find the best configuration for specific tasks. The complexity of the RNP framework, particularly with the incorporation of robust divergences, can make it harder to interpret the model's decisions and the underlying relationships in the data. This lack of interpretability may hinder its adoption in fields where understanding model behavior is crucial, such as healthcare or finance.

**Questions:**

1, How does the choice of Renyi divergence impact the trade-off between predictive performance and computational efficiency in RNP compared to traditional Neural Processes?

2, What strategies can be employed to effectively tune the additional parameters introduced in RNP to ensure optimal model performance across different applications?

---

> ### Author Response · Authors · 2024-11-22
>
> # MC efficiency.
> We would like to clarify to the reviewer that our RNP framework *consumes the same resource as the standard NPs* since they usually require multiple samples to better estimate the likelihood model as well. Our overhead computational cost comes from non-analytical solutions for the divergence, whereas our ablations in Fig 5 showed that it takes as few as 8 samples during training to gain better estimates, and one can choose to increase the number of samples during inference for better predictive performance or to reduce it for scalability and real-time applications. We also reported the wall clock time between standard NPs and RNPs in Table 6 (Supplementary) and no significant differences were found.
>
> # Hyperparameter sensitivity and automatic tuning.
> Please refer to the hyperparameter sensitivity comment for Reviewer T3rx.
>
> # Interpretation with robust divergence.
>
> Could the reviewer clarify further about the interprebtability aspect of the model? As far as we understand, neither neural processes nor general neural networks focus on effective interpretability, and RNPs already facilitate better decision makings by providing both the data uncertainty estimation $p(y|\mathbf{x}, C)$ and functional uncertainty estimation $p(\mathbf{z}|{C})$. Regarding relevant healthcare and finance applications, please check the related work [3] where NPs were validated on covid case forecasts and stock pricing prediction (Fig 4 and Fig 8 in [3]).
>
> [3] Wang, Xuesong, et al. "Global convolutional neural processes." 2021 IEEE International Conference on Data Mining (ICDM). IEEE, 2021.
>
>
> # Question 1 RNP efficiency.
>
>  One can trade-off between performance and efficiency by increasing or reducing the number of samples in MC for training and inference. In cases where posterior distributions are not standard Gaussians, our framework is equally efficient as the standard NPs.

---

> > ### Author Response · Authors · 2024-11-25
> >
> > Dear Reviewer,
> >
> > Hopefully our responses have helped address your concerns. We are more than happy to provide further discussions if needed.
> >
> > Regards

---

> > > ### Comment · Reviewer_eR5B · 2024-11-26
> > >
> > > I thank the authors for their rebuttal, though it was not sufficiently convincing to address my concerns. For example, "neither neural processes nor general neural networks focus on effective interpretability". Please check the literature: there are tons of papers (e.g., XAI) trying to explain neural network decisions. Therefore, I decided to maintain my previous score.

---

> > > > ### Author Response · Authors · 2024-11-26
> > > > **Please let us know more specific feedback**
> > > >
> > > > We would very much appreciate if the reviewer, besides the interpretability issue, could let us know why our response is not "sufficiently convincing", as we believe we have provided enough arguments wrt efficiency and hyper-parameter sensitivity. knowing the precise arguments for this would allow for a healthier academic discussion.
> > > >
> > > > With regards to interpretability, we quote the reviewers' original comment
> > > > >> The complexity of the RNP framework, particularly with the incorporation of robust divergences, can make it harder to interpret the model's decisions and the underlying relationships in the data. This lack of interpretability may hinder its adoption in fields where understanding model behavior is crucial, such as healthcare or finance.
> > > >
> > > > To the best of our knowledge, neither the original NP framework or the subsequent NP developments are readily interpretable. Is the reviewer suggesting otherwise? While we agree with the reviewer that "there are tons of papers [on interpretable models] (e.g., XAI)", this is not the focus of our work and it is an orthogonal direction of research. We believe this concern is not very relevant to our contribution and certainly out of scope, although interesting for future work.

---

> ### Author Response · Authors · 2024-11-28
>
> Dear Reviewer ,
>
> We greatly appreciate your time and effort in reviewing our work.
>
> Regarding the reviewer's initial concern regarding **hyperparameter sensitivity and automatic tuning**, please check the heuristic (3) we left for the Reviewer T3rx:
>
> To start with standard KL objective ($\alpha \rightarrow$ 1) and gradually decrease $\alpha$ (with granularity according to computational constraints) to 0. The intuition is inspired by KL annealing for VAE models, which starts with a strong prior penalization ($\alpha$ close to 1) to reduce the posterior variance quickly and gradually reduces the prior penalization ($\alpha$ close to 0) and focuses more on model expressiveness.
>
> We have **just added the experimental results in Supplementary table 5 to support this strategy, which is more efficient and saves the computational cost of hyperparameter search.**
>
>
>
> Since the author-reviewer discussion deadline is fast approaching, we kindly ask for feedback on our responses.
>
>  We would also like to confirm whether our responses have adequately addressed your initial concerns. If so, we kindly ask **whether you might reconsider your original score**.
>
> Best regards,
>
> The Authors

---

> > ### Author Response · Authors · 2024-12-03
> >
> > Dear Reviewer eR5B,
> >
> > We hope this message finds you well. We truly appreciate your engagement with our rebuttal and thank you for your insightful comments and questions.
> >
> > We have provided further clarification to address your questions. As the rebuttal discussion period ends soon, we would be grateful for your feedback on whether our responses have adequately addressed your concerns. We are ready to answer any further questions you may have.
> >
> > Thank you for your valuable time and effort!
> >
> > Best regards,
> >
> > The Authors

---

### Official Review · Reviewer_T3rx · 2024-11-03

**Soundness:** 3
**Presentation:** 3
**Contribution:** 2
**Rating:** 3
**Confidence:** 4

**Summary:**

The paper proposes RNPs, a new framework that aims to mitigate the issue of prior misspecification in neural processes (NPs). NPs are deep probabilistic models that represent stochastic processes by conditioning prior distributions on context points. However, the parameterization coupling between the prior and posterior models in NPs can lead to a misspecified prior, resulting in biased posterior estimates and degraded performance.

To address this, the authors propose optimizing an alternative posterior using the Rényi divergence between the model posterior and the true posterior, instead of the standard KL divergence. The Rényi divergence introduces a hyperparameter α that scales the density ratio between the posterior and prior, dampening the effects of the misspecified prior. The proposed RNP objective unifies the variational inference and maximum likelihood estimation objectives for training NPs via α, allowing better marginal likelihood and posterior expressiveness.

**Strengths:**

- The paper provides a thorough theoretical analysis and derivations for the proposed Rényi Neural Process objective, establishing a solid foundation for the proposed method.
- The experiments are comprehensive, covering various datasets and tasks, including regression, image inpainting, and real-world regression problems with prior misspecification.
- The authors conduct extensive ablation studies and hyperparameter tuning, demonstrating the robustness and effectiveness of the proposed method.

**Weaknesses:**

- Limited novelty: The core idea of using an alternative divergence measure (Rényi divergence) to mitigate the effects of prior misspecification is not entirely novel. Previous works in the domain of robust variational inference have explored the use of other divergences, such as α-divergences and f-divergences, to address similar issues. The authors could provide a more comprehensive discussion of how their work relates to and differentiates from these earlier efforts.
- Hyperparameter sensitivity: The choice of the hyperparameter α plays a crucial role in the performance of the proposed method. While the authors provide some guidelines for tuning α, a more comprehensive analysis of the sensitivity of the method to different values of α and strategies for automatic tuning could further enhance the practical utility of the proposed framework.

**Questions:**

- The authors mention that the proposed framework can be further extended to improve the robustness of contextual inferences, such as prompt design in large language models. Could the authors provide more details or insights into how their method could be adapted or applied to such tasks?

- The parameter coupling between the posterior and prior in neural processes is by design to share parameters, but it's not a hard constraint. Isn't a simple baseline approach to mitigate prior misspecification to separately parameterize the posterior and prior models? Could the authors provide a comparison with this simple baseline to better demonstrate the advantages of their proposed method?

Overall, naively combining existing ideas may not constitute a sufficient contribution for a top-tier conference like ICLR, which expects a higher level of novelty and significance.

---

> ### Author Response · Authors · 2024-11-22
> **Part 1 of 2**
>
> # Limited novelty.
>
> We would like to clarify to the reviewer that our work goes well beyond using a different divergence for variational inference in neural processes (NPs). Firstly, we are the first to identify that the parameter coupling between the prior conditional and the approximate posterior inherent to NPs amounts to prior misspecification. Secondly, we introduce a new objective that addresses this misspecification and that unifies the variational inference (VI) and maximum likelihood estimation (MLE) objectives. Finally, besides developing the supporting theory in section 3,  we show that our RNP method can be applied to several SOTA NP families *without changing the models*, and provides significant generalization performance improvements over competing approaches.
>
>
> # Hyperparameter sensitivity.
>
> We would like to emphasize that we have carried out a sensitivity analysis on the hyperparameter $\alpha$ in the main paper. The main conclusion across all datasets and models is that $\alpha=0.7$ provides significant performance improvements over the competing approaches. Hence, we recommend this value as default to practitioners and researchers. (See also the additional clarification in section 7.1.) One may also use the prior knowledge to select the $\alpha$ based on the understanding of the misspecification. Nevertheless, we recognize that in other scenarios such as very different datasets and/or models, this default value may not work as well as in our experiments. For this purpose, we have found that cross-validation is an effective tool for finding near-optimal values.
> Since cross-validation is computationally expensive, we suggest the following heuristics:
>
>
>  1. Start with a value close to 1, which corresponds to standard KL minimization.
>
>  2. Only consider $0 < \alpha < 1$ since otherwise we may violate the conditions for the divergence between two Gaussians.
>
>  3. Gradually decrease $\alpha$ (with granularity according to computational constraints) to 0. The intuition is inspired by KL annealing for VAE models [1], which starts with a strong prior penalization ($\alpha$ close to 1) to reduce the posterior variance quickly and gradually reduces the prior penalization ($\alpha$ close to 0) and focuses more on model expressiveness.
>
>     [1] Bowman, Samuel, et al. "Generating Sentences from a Continuous Space." Proceedings of the 20th SIGNLL Conference on Computational Natural Language Learning. 2016.
>
> # LLM application.
>
> One can formulate the problem of prompt design for vision-language models as maximizing a contextual probability $\mathbb{C}: \max_{\mathbb{C}} p_{\mathbb{C}}(y| \mathbf{x}, \mathbb{C})$ (Eq (3) in [2]) where $\mathbf{x}$ is an image, $y$ is its corresponding class, e.g., "dog", and $\mathbb{C} \in \mathbb{R}^{n_c \times d_c}$ is a sequence of prompt embeddings to improve the predictive likelihood, e.g., the embedding of "a photo of an [Object]". Similar to our maximum likelihood formulation in Eq (8), it can be viewed as an NP problem, and the parameters $\theta$ of in Eq(8) becomes the context set $\mathbb{C}$ itself. Therefore, the training can facilitate our Renyi objective in Eq (12) to make better inference.
>
> [2] Zhou, Kaiyang, et al. "Learning to prompt for vision-language models." International Journal of Computer Vision 130.9 (2022): 2337-2348.

---

> > ### Author Response · Authors · 2024-11-22
> > **Part 2 of 2**
> >
> > # Different prior and posterior parameterization.
> >
> > One of our baseline models VNP [3] adopted different prior and posterior parameterization (See Eq (7) and Eq (8) in the paper) and our RNP objective still showed improvements across comprehensive datasets in Table 1. We have clarified this setting in the paper.
> >
> > We also compared the RNP with the separate prior-posterior parameterization for NPs and ANPs(which explicitly defined the prior models). The results in Table 4 (Supplementary) showed that RNP still outperformed this baseline with much fewer parameters. The theoretical explanation for the underperformance of the baseline could be the violation of the consistency property of NPs. Take a special case where the target set $(X_T, Y_T)$ is a subset of the context set $(X_C, Y_C)$, the NP objective Eq (1) using separate parameterization becomes
> > $\mathcal{L}(\phi, \theta, \varphi)  = -\mathbb{E}_{q(\mathbf{z}| X_T, Y_T, C, \phi)} \log p(Y_T|X_T, \mathbf{z}, \theta) + {KL}\left(q(\mathbf{z}| X_T, Y_T,  C, \phi) \mid p(\mathbf{z}|C, \varphi)\right) $.
> >
> > Different parameterizations of $\varphi$ and $\phi$ make the second KL term non-zero and the marginal is not consistent anymore: $p(Y_T|X_T, C) = \int p(Y_T|\mathbf{z}, X_T) q(z|{C, \varphi}) d\mathbf{z} \neq \int p(Y_T, Y_C|X_T, {C})dY_C = \int p(Y_T| \mathbf{z}, X_T)p(Y_C|\mathbf{z}, X_C) q(\mathbf{z}| X_T, Y_T, \mathbb{C}, \phi)d\mathbf{z}d{Y_C}$,
> >  hence the corresponding NPs can no long represent a stochastic process.
> >
> > [3]. Guo, Zongyu, et al. "Versatile Neural Processes for Learning Implicit Neural Representations." The Eleventh International Conference on Learning Representations (ICLR), 2023.

---

> > > ### Comment · Reviewer_T3rx · 2024-11-24
> > >
> > > Thanks for the clarification. I have several more questions:
> > >
> > > 1. regarding LLM prompt optimization. I understand Eq 8 can be re-formulated to optimize the context C, but how can we use Eq12 is unclear to me. All those LLM/VLM have specific architectures, what is the latent code $z$ for them?
> > >
> > > 2. > Different parameterizations of $\varphi$ and $\phi$ make the second KL term non-zero and the marginal is not consistent anymore
> > >
> > > Even you use the same parametrization, it's not guaranteed to be consistent, right?

---

> ### Author Response · Authors · 2024-11-24
>
> We thank the reviewer for the prompt response.
>
> # Prompt optimization.
>
> Although different LLMs/VLMs have specific architectures, our goal is to learn the prompt embedding $\mathbf{C} \in \mathbf{R}^{nc \times dc} $ which *does not depend on the architectures* and therefore can be utilized in a general formulation. Recall in the reference [2] the authors also highlighted that ``CoOP models a prompt’s context words with learnable vectors while the entire pre-trained parameters are kept fixed''.  In terms of introducing the latent code $\mathbf{z}$, please refer to our eq (11),(12) and the reparameterization trick introduced in the paper:
>
> $ \mathbf{z}_k = s(\mathbf{C}, \epsilon_k, \varphi), \epsilon_k \sim \mathcal{N}(\mathbf{0}, I)$
>
> where $s$ is a neural network parameterized by $\varphi$ that takes the context embedding $\mathbf{C}$ and a random noise $\epsilon_k$ as inputs, where both the context embedding $\mathbf{C}$ and the $\varphi$ are learnable parameters. The intuition behind it is instead of modeling the deterministic prompt embedding, we can model its distribution.
>
> # Stochastic process consistency.
> In the example we presented where the target set is assumed as a *subset* of the context set, it is consistent under the same parameterization of $\phi$ and $\varphi$. We show the proof with the unified $\varphi$ and $\phi$ here:
>
>  $\mathcal{L}(\phi, \theta)  = -\mathbb{E}_{q(\mathbf{z}| X_T, Y_T, C, \phi)} \log p(Y_T|X_T, \mathbf{z}, \theta) + {KL}\left(q(\mathbf{z}| X_T, Y_T,  C, \phi) \mid q(\mathbf{z}|C, \phi)\right) $
>
> $ =  -\mathbb{E}_{q(\mathbf{z}| C, \phi)} \log p(Y_T|X_T, \mathbf{z}, \theta) + {KL}\left(q(\mathbf{z}|C, \phi) \mid q(\mathbf{z}|C, \phi)\right)$.
>
> The KL term becomes 0 because {$X_T, Y_T, \mathbf{C}$} = $\mathbf{C}$ and hence,  the marginal is consistent with the marginalization of the joint $p(Y_T|X_T, C) = \int p(Y_T|\mathbf{z}, X_T) q(z|{C, \phi}) d\mathbf{z} = \int p(Y_T, Y_C|X_T, {C})dY_C = \int p(Y_T| \mathbf{z}, X_T)p(Y_C|\mathbf{z}, X_C) q(\mathbf{z}| X_T, Y_T, \mathbf{C}, \phi)d\mathbf{z}d{Y_C} = \int p(Y_T| \mathbf{z}, X_T)p(Y_C|\mathbf{z}, X_C) q(\mathbf{z}| \mathbf{C}, \phi)d\mathbf{z}d{Y_C} $.
>
> Hopefully our responses have helped address your concerns. We are more than happy to provide further discussions if needed.

---

> > ### Comment · Reviewer_T3rx · 2024-11-26
> >
> > Thanks for the clarification. The consistency only holds for this special case, where target set is a subset of the context set. But in reality, it's usually the opposite, where context set is a subset of target set. This is probably off-topic and it's a flaw of all NP models using variational approximation, as the posterior approximation gap always exists.

---

> > > ### Author Response · Authors · 2024-11-27
> > >
> > > Thank you for your additional remarks, which we will incorporate into the final version. We would also like to confirm whether our responses have adequately addressed your initial concerns. If so, we kindly ask whether you might reconsider your original score.

---

> ### Author Response · Authors · 2024-11-28
> **Additional results for $\alpha$ tuning and discussion**
>
> We greatly appreciate the reviewer’s valuable feedback, which has been very helpful throughout this process. We have **revisited** the initial comments of the reviewer and we genuinely hope that our further clarifications can address some of your concerns.
>
> 1. Regarding the reviewer's initial concern for the **automatic tuning of $\alpha$**, we have previously provided a heuristics.  Here, we provide the experimental evidence by comparing with the standard NP objectives. Our heuristics still outperform the baselines across multiple datasets and methods.
>
> | Method | Set     | Objective   | RBF        | Matern 5/2 | Periodic   | MNIST      | SVHN        |   |   |   |   |   |   |   |
> |--------|---------|-------------|------------|------------|------------|------------|-------------|---|---|---|---|---|---|---|
> | NP     | context | $\mathcal{L}_{VI}$        |  0.69±0.01 |  0.56±0.02 | **-0.49±0.01** |  0.99±0.01 |  3.24±0.02  |   |   |   |   |   |   |   |
> |        |         | $\mathcal{L}_{RNP+Ada\alpha}$ |  **0.75±0.02** |  **0.61±0.02** | **-0.49±0.00** |  **1.01±0.01** |  **3.26±0.01**  |   |   |   |   |   |   |   |
> |        | target  | $\mathcal{L}_{VI}$        |  0.26±0.01 |  0.09±0.02 | **-0.61±0.00** |  0.90±0.01 |  3.08±0.01  |   |   |   |   |   |   |   |
> |        |         | $\mathcal{L}_{RNP+Ada\alpha}$ |  **0.31±0.01** |  **0.13±0.01** | **-0.61±0.00** |  **0.92±0.01** |  **3.10±0.01**  |   |   |   |   |   |   |   |
> | ANP    | context | $\mathcal{L}_{VI}$        |  **1.38±0.00** |  **1.38±0.00** |  0.65±0.04 |  **1.38±0.00** |  **4.14±0.00**  |   |   |   |   |   |   |   |
> |        |         | $\mathcal{L}_{RNP+Ada\alpha}$ |  **1.38±0.00** |  **1.38±0.00** |  **0.97±0.11** |  **1.38±0.00** |  **4.14±0.00**  |   |   |   |   |   |   |   |
> |        | target  | $\mathcal{L}_{VI}$     |  0.81±0.00 |  0.64±0.00 | -0.91±0.02 |  **1.06±0.01** |  **3.65±0.01**  |   |   |   |   |   |   |   |
> |        |         | $\mathcal{L}_{RNP+Ada\alpha}$ |  **0.83±0.01** |  **0.66±0.01** | **-0.71±0.05** |  **1.06±0.01** |  **3.65±0.01**  |   |   |   |   |   |   |   |
> | TNP-D  | context |$\mathcal{L}_{ML}$      |  **2.58±0.01** |  **2.57±0.01** | **-0.52±0.00** |  1.73±0.11 |  10.63±0.12 |   |   |   |   |   |   |   |
> |        |         | $\mathcal{L}_{RNP+Ada\alpha}$ |  **2.58±0.01** |  **2.57±0.01** | **-0.52±0.00** |     **1.94±0.02**        |     **10.73±0.57**         |   |   |   |   |   |   |   |
> |        | target  | $\mathcal{L}_{ML}$         |  1.38±0.01 |  **1.03±0.00** | **-0.59±0.00** |  **1.63±0.07** |   6.69±0.04 |   |   |   |   |   |   |   |
> |        |         | $\mathcal{L}_{RNP+Ada\alpha}$ |  **1.39±0.00** |  **1.03±0.00** | **-0.59±0.00**  |     1.56±0.02        |      **6.71±0.24**         |   |   |   |   |   |   |   |
>
>
> 2. **Relationship with robust divergences**.
>
> We have elaborated the relationships with respect to robust divergences in the related work section. We have discussed certain limitations of the robust divergences,  e.g., the specifications of the convex function f and deriving its dual form for f-divergence VIs [1].  In terms of their relationships with our work, "As far as we understand, we are the first to analyse the limitations of NPs from the perspective of prior misspecifications, and facilitate robust divergence to enable better NP learning."
>
> [1] Wan, Neng, Dapeng Li, and Naira Hovakimyan. "F-divergence variational inference." Advances in neural information processing systems 33 (2020): 17370-17379.
>
>
> We are eager to ensure that we have adequately addressed your concerns and are prepared to offer further clarifications or address any additional questions you may have. We look forward to hearing your feedback.

---

> > ### Author Response · Authors · 2024-12-03
> >
> > Dear Reviewer T3rx,
> >
> > We hope this message finds you well. We truly appreciate your engagement with our rebuttal and thank you for your insightful comments and questions.
> >
> > We have provided further clarification to address your questions. As the rebuttal discussion period ends soon, we would be grateful for your feedback on whether our responses have adequately addressed your concerns. We are ready to answer any further questions you may have.
> >
> > Thank you for your valuable time and effort!
> >
> > Best regards,
> >
> > The Authors

---

### Author Response · Authors · 2024-11-22
**Summary**

We would like to thank all the reviewers for their constructive feedback and the time and effort applied in reviewing our manuscript. We provide individual responses to each review, but we also address some of the main common points here.

# Motivation of RNPs and analytical analysis of prior misspecifications.

 We analytically compared the gradients of our RNP objective with the gradients using the VI objective to show how $\alpha$ would affect the posterior parameter updates.

 # Sensitivity analysis and tuning of the hyperparameter $\alpha$.

 We would like to emphasize that we have carried out a sensitivity analysis on the hyperparameter $\alpha$ in the main paper. The main conclusion across all datasets and models is that $\alpha=0.7$ provides significant performance improvements over the competing approaches. Hence, we recommend this value as default to practitioners and researchers. (See also the additional clarification in section 7.1.) Nevertheless, we recognize that in other scenarios such as very different datasets and/or models, this default value may not work as well as in our experiments. For this purpose, we have found that cross-validation is an effective tool for finding near-optimal values. Since cross-validation is computationally expensive, we have also suggested heuristics to set $\alpha$. See below for details.

 # MC sampling efficiency.

 We would like to clarify that our RNP framework *consumes the same resource as the standard NPs* since they usually require multiple samples to better estimate the likelihood model as well. Our computational overhead comes from non-analytical solutions for the divergence, whereas our ablations in Fig 5 showed that it takes as few as eight samples during training to gain better estimates, and one can choose to increase the number of samples during inference for better predictive performance or to reduce it for scalability and real-time applications.

  # Changes in the paper.

 We have crystallized our contributions in section 1. We have improved clarifications on prior misspecifications and the reparameterization tricks used for the VI-based objectives in section 3.1. We have revised the formulation of the ML-objective and in section 3.2 and the gradient update in section 4. More experimental setups are elaborated on the prior misspecification datasets in section 7.2. We have fixed typos throughout the paper and polished the plots with better resolutions and bigger legends.

 # Adding new results.

We have added two tables in the rebuttal to compare our RNP with a baseline model suggested by Reviewer T3rx (Supp Table 4)  and to compare MC efficiency suggested by Reviewer eR5B via wall clock time comparison between RNP and standard Neural processes (Supp Table 5).

---

### Meta-Review · Area_Chair_RNd7 · 2024-12-26

**Metareview:**

The  paper introduces Neural Renyi processes, where the neural processes objectives are revisited with Renyi divergences. Authors claim that the Renyi losses overcome prior misspecification, i,e when the target distribution does not belong to the parametrized family , i,e an identifiability problem.

Reviewers raised several concerns, one regarding interpretability (Reviewer eR5B), and another one on the link between the prior mispecification and Renyi divergence (Reviewer 6B6t), and lack of novelty (Reviewer T3rx). Reviewer JR2F provided a detailed review that has improved the paper greatly.

Given the large variance in the scores AC and reviewers discussed the paper, and I read the paper in details. I am not concerned with novelty or with the interpretability of the method. One thing I am still missing as I read the paper is how the prior misspecification is linked  to the use of Renyi divergence.

The authors should provide a theoretical insight about this claim their work falls within the framework of [a], as it amounts to an importance sampling , this is largely studied in this paper theoretically.  The other aspect is that Renyi divergence minimization is rather a great alternative to KL when the target is heavy tailed or does not have defined moments see [b]. A rigorous proof for Renyi minimization alleviating the misspecification prior would strengthen the paper, or in terms of better targeting the tails as in [b] would bridge more the algorithmic part and the motivation of the work.


[a] https://www.jmlr.org/papers/volume24/22-1160/22-1160.pdf

[b] https://hal.science/hal-04506761/document

**Additional Comments On Reviewer Discussion:**

This is a borderline paper, the paper was discussed between authors and reviewers, and reviewers and AC. Given the discrepancy in the scores the decision was made based on the the reviewers comments on the lack of clarity in the paper and the disconnect between  motivation of the work and the proposed method.

---

### Decision · Program_Chairs · 2025-01-22

Reject